# Robust immune response to COVID-19 vaccination in the island population of Greenland
Mie Møller [1,2,3,4,5] ✉, Lennart Friis-Hansen [6,7], Nikolai Kirkby[6], Christine Dilling-Hansen[6], Mikael Andersson [8], Peter Vedsted[9,10], Kåre Mølbak [2,4] & Anders Koch[1,3,4,5]

## Abstract

**Background** In Greenland, the COVID-19 pandemic was characterised by a late onset of community transmission and a low impact on the healthcare system, hypothesised as being partly due to a high uptake of vaccinations. To underpin this description, we aimed to assess the SARS-CoV-2 immune response post-vaccination in a Greenlandic population. **Methods** In this observational cohort study, we included 430 adults in Greenland who had received a complete two-dose SARS-CoV-2 vaccination at enrolment. The total plasma SARS-CoV-2 spike glycoprotein Ig antibodies (S-Ab) induced by either the BNT162b2 or mRNA-1273 vaccine, was measured up to 11 months after the second vaccine dose. In addition, total salivary S-Abs were examined in 107 participants, and the T-cell response to the spike glycoprotein was assessed in 78 participants out of the entire study cohort. **Results** Here we demonstrate that two months after the second vaccine dose, 96% of participants have protective plasma S-Ab levels. By 11 months, 98% have protective levels, with prior SARS-CoV-2 infection particularly enhancing S-Ab levels by 37% (95% CI 25–51%). Among individuals aged 60 years and older, we observe a 21% (95% CI 7–33%) reduction in antibody response. Total salivary S-Ab levels are detectable in all participants and significantly correlate with plasma levels. Moreover, all participants exhibit a robust SARS-CoV-2-specific T-cell response 11 months post-primary vaccination. **Conclusions** Our findings show that Greenlanders exhibit a robust and lasting immune response, both humoral and cellular, comparable to other population groups up to at least 11 months after the second vaccine dose. These results corroborate the hypothesis that vaccines contributed to the mild impact of the COVID-19 pandemic in the Greenlandic population.

## Plain Language Summary

Effective public health measures were crucial to protect Greenland's vulnerable population against the COVID-19 pandemic. Vaccines were particularly important, although their effectiveness in Greenland's unique and isolated population had not been explored. Our aim was to determine the COVID-19 vaccines' immunological response as a measure of protection among Greenlanders. By measuring antibody levels and immune cell activity, we demonstrate that over nine out of ten Greenlanders remained well protected by COVID-19 vaccines up to 11 months after their second vaccine dose, although older adults were less well protected. Prior COVID-19 infection or a booster dose enhanced protection against severe disease. These findings provide valuable insights for Greenland and similar ancestral and geographical populations, aiding in their ongoing vaccination strategies and future pandemic preparedness.

The Severe Acute Respiratory Syndrome Coronavirus 2 (SARS-CoV-2) emerged in December 2019, resulting in a global pandemic with a high number of fatalities[1]. Rapid development and deployment of vaccines played a crucial role in reducing severe illness and deaths caused by the SARS-CoV-2 virus worldwide[2].

While most countries had to abandon a containment strategy by the spring of 2020, the course of the epidemic in Greenland was substantially different as community transmission was delayed until the autumn of 2021, when vaccination uptake was high. Travel and domestic restrictions played important roles in the containment, but other aspects of the COVID-19 epidemic in Greenland are not well documented[3].

[1]Institue of Health and Nature, University of Greenland, Nuuk, Greenland. [2]Department of Veterinary and Animal Sciences, University of Copenhagen, Copenhagen, Denmark. [3]Department of Internal Medicine, Queen Ingrid's Hospital, Nuuk, Greenland. [4]Department of Infectious Disease Epidemiology and Prevention, Statens Serum Institut, Copenhagen, Denmark. [5]Department of Infectious Diseases, Rigshospitalet University Hospital, Copenhagen, Denmark. [6]Department of Clinical Microbiology, Rigshospitalet University Hospital, Copenhagen, Denmark. [7]Department of Clinical Biochemistry, Bispebjerg University Hospital, Copenhagen, Denmark. [8]Department of Epidemiology Research, Statens Serum Institut, Copenhagen, Denmark. [9]Department of Clinical Medicine, University of Aarhus, Aarhus, Denmark. [10]Ilulissat Regional Hospital, Ilulissat, Greenland. ✉e-mail: mimoe@uni.gl

In Greenland, the mRNA-1273 (Moderna) vaccine was administered to the majority of the population, while a smaller segment, including the elderly and critical staff e.g., healthcare professionals, received the BNT162b2 (Pfizer/BioNTech) vaccine. Vaccine distribution was challenging due to the scattered population in coastal towns and settlements. Healthcare workers transported vaccines in large freezers on ships to reach remote communities, and in northern regions, some settlements were only accessible by dog sled. However, by June 2022, 66% of the population had received an initial two-dose vaccine regime, and 41% had received a third booster dose[3]. Despite good vaccination coverage compared to other Arctic regions[4], over the course of the pandemic, Greenland experienced a considerable number of COVID-19 cases (21,419/100,000 inhabitants as of June 2022). However, although some cases (2442/100,000 inhabitants as of November 2021) occurred before the advent of the Omicron (B.1.1.529) variant, generalised community transmission was not seen until this variant started to become dominant by late 2021[3].

The population of Greenland is characterised by high rates of respiratory (including tuberculosis) and chronic disease including obesity[5], and inadequate housing (i.e., overcrowding). Additionally, logistical barriers to accessing health services exist[6], such as large distances that can only be covered by flight or boat, limited availability of hospitals in remote regions, lack of healthcare professionals, and specialised equipment like ventilators. Except for the main hospital in the capital Nuuk, all other hospitals are small, and mainly staffed by general practitioners and junior doctors. Therefore, from the onset of the pandemic, the Greenlandic health authorities expected COVID-19 to represent a major burden to the health of the Greenlandic population. To minimise the import of SARS-CoV-2 to Greenland, strict travel restrictions such as complete border closure, mandatory quarantine, and testing of all travellers entering Greenland were quickly implemented by the Greenlandic authorities. These measures aimed at delaying the peak of transmission, providing time for the implementation of additional public health interventions, including vaccines. The relatively low numbers of COVID-19 related hospitalisations (27 individuals) and deaths (11 individuals) as of June 2022 suggest, among other mitigating factors, the protective value of vaccine immunisation before the occurrence of community transmission[3,7].

A number of Danish studies have examined the effectiveness of COVID-19 vaccines[8,9]. These vaccines were found to induce robust humoral and cellular responses depending on the vaccine type, booster vaccination, previous SARS-CoV-2 infection, or other factors e.g., age. However, evaluation of vaccine effects in the Greenlandic (part of the Kingdom of Denmark) or other Inuit populations has not yet been conducted. Some studies suggest that the immunogenicity of COVID-19 vaccines can be influenced by ethnicity[10]. Given that previous research has shown differences in vaccine response between Inuit populations and other ethnic groups for a range of other infections such as measles and Haemophilus influenzae type b[11,12], it was important to investigate the influence of ethnicity particularly within the Inuit population, on the response to the COVID-19 vaccines. The immunogenicity of vaccines in Inuit populations is not well explored and the reasons for different vaccine responses among Inuits and non-Inuits are unclear[11]. However, these previous studies highlight the need for immunogenicity studies on indigenous populations before generalising the results of vaccine trials done in predominantly Caucasian populations[11]. Furthermore, determining the dynamics of vaccine-induced immune responses in the Greenlandic population is important for understanding the pandemic's trajectory and guiding future vaccination strategies in Greenland and similar isolated areas.

To assess the long-term vaccine-induced antibody-mediated immune response against SARS-CoV-2 in a Greenlandic population, we conduct a cohort study, measuring the humoral and cellular antibody responses over 11 months after primary vaccination. Specific aims are to quantify the COVID-19 vaccine-induced immune response by analysing the total SARS-CoV-2 spike glycoprotein Ig antibody levels (S-Ab) in plasma and saliva, and the T-cell response to the spike glycoprotein in plasma.

## Methods

### Study design

The study was designed as a cohort study of adults, born and living in Greenland. The study population included individuals who had received a two-dose regime with the mRNA-1273 (Moderna) or the BNT162b2 (Pfizer/BioNTech) vaccine ~2 months before the time of enrolment.

### Setting

Greenland is an island of 2,166,086 km², where 81% of the country is covered by the icecap. It has a population of 56,583 individuals (89% of Inuit descent[3]) as of January 2021[13], with 60 settlements and 17 towns scattered along the coastline. The main transportation between major towns is by ship or plane as there are no roads connecting towns. The majority of Greenland's population resides along the southwest coast, including the capital Nuuk[3,13]. Greenland is an autonomous country within the Kingdom of Denmark and operates its own tax-financed healthcare system. Greenland is divided into five healthcare regions, each hosting a regional hospital in the main town of the given region. Other towns in each region have healthcare centres, while settlements are equipped with healthcare stations with or without health-educated staff (MDs, nurses, nurse practitioners, midwives, healthcare assistants, and health aides). The Queen Ingrid's Hospital in Nuuk serves as the national referral hospital for all of Greenland and is the sole facility with an intensive care unit in the country.

### Study population and data collection

The eligible study population was adults 18 years of age or older, who were born in Greenland and living in one of the two major towns on the Greenlandic Westcoast; the capital Nuuk (population 19,604[13]) or the town of Ilulissat (600 km north of Nuuk, population 4670[13]). All participants had received a two-dose vaccination regimen (considered as primary vaccination) with either the mRNA-1273 (Moderna) vaccine or the BNT162b2 (Pfizer/BioNTech) vaccine approximately two months before enrolment.

At enrolment, all participants filled in a questionnaire in Greenlandic or Danish with their personal information (name, date of birth, civil registration number, residency, place of birth, and place of birth of parents), comorbidities, and daily medications. Medical history and information on primary vaccination dates and types were collected and confirmed through the Greenlandic Electronic Medical Patient Registry COSMIC.

Based on the places of birth of parents, participants were classified as either Inuit (both parents born in Greenland), mixed (one parent born in Greenland), or non-Inuit (no parent born in Greenland).

At enrolment and the 11-month follow-up visit, participants were asked to give information (test result(s) and date(s) of testing) of possible previous SARS-CoV-2 infection(s), possible symptoms, severity (i.e., hospitalisation) and/or booster vaccination status. A breakthrough infection was defined as a positive test for SARS-CoV-2 after two doses or more using either a PCR test or a rapid antigen test. Reinfection was defined as having a positive test more than 90 days after the last positive test. Due to limited resources, not all positive COVID tests were recorded in the National Health Registries, making this information reliant on self-reporting. Additionally, some participants might have received a booster vaccination in Denmark, which was not registered in the Greenlandic Health Registries. Therefore, information concerning booster vaccinations was also based on self-reports.

### Sample collection and handling

The study collected blood samples in all participants ~2 months and 11 months after the second vaccine dose to assess the vaccine-induced antibody immune response (Supplementary Fig. 1). Due to logistical limitations, it was not feasible to conduct additional sample collections despite the extended time interval between these points. Additionally, test kits for measurement of salivary antibody and T-cell responses were not suitable for research in remote locations with limited laboratory facilities like Greenland at the first round of sampling. Therefore, we only collected saliva samples to determine the salivary antibody response and additional blood samples to determine the general and specific SARS-CoV-2 T-cell response from a

minor subgroup of the included participants at 11 months after the second vaccine dose.

For antibody measurements, 3 mL of blood was collected in Vacutainer EDTA tubes (Becton Dickinson, Plymouth, United Kingdom, cat no. 368857). The samples were centrifuged within 2 h (4000 $g$ for 10 min), and plasma was collected, frozen, and kept at $-20\,°C$. Saliva samples were collected using the Salivette® Cortisol collection tubes (Sarstedt, Nümbrecht Germany, cat no. 51.1534.5008). Briefly, the person held a synthetic swab in the mouth for 2 min. Afterwards, the swab was placed in a collection container and centrifuged (3100 $g$ for 5 min) to remove mucus and other contaminants, after which ~1 mL of clear oral fluid was collected, frozen, and kept at $-20\,°C$.

T-cell responses were measured using interferon-γ release assays (IGRAs). Blood samples were collected in a 6 mL heparin tube (Becton Dickinson, cat no. 368886). Within 2 h after collection, 1 mL whole blood was transferred to each of the four QuantiFERON® SARS-CoV-2 tubes (Starter Pack, cat no./ID: 626715) to measure the specific T-cell response to the SARS-CoV-2 spike glycoprotein. Additionally, 1 mL of whole blood was transferred to the QuantiFERON monitor (QFM) tube (Qiagen®, Hilden, Germany, cat no./ID: 0650-0101) to measure the general T-cell response. After thorough mixing, the tubes were incubated at $+37\,°C$ for 22–24 h, after which they were centrifuged (3300 $g$ for 15 min) to separate plasma from blood cells.

All samples were frozen ($-20\,°C$) within 4 h of processing and shipped frozen to Denmark for final laboratory analysis.

## Laboratory tests
Total (IgG+A+M) SARS-CoV-2 spike glycoprotein antibodies (S-Abs) and total nucleoprotein antibodies (N-Abs) were measured using ECLIA Assays (Roche Diagnostics, Mannheim, Germany)[14]. The N-Ab cut-off for positivity was defined as >0.8 COI (cut-off index)[15]. The S-Ab assay had a measuring range of 0.4–25,000 kBAU/L (kilo-binding antibody units per litre) and a cut-off for positivity >0.8 KBAU/L. Antibody levels below 300 kBAU/L were considered insufficient (non-protective) as a response to the index variant[16,17].

T-cell responses were evaluated using the QuantiFERON® SARS-CoV-2 and the QFM assays which are IGRAs. IGRAs assess T-cell responses by quantifying the amount of IFN-γ released after 24 h of incubation of a fixed amount of whole stimulated peptides, small molecules, or antibodies. The released IFN-γ concentration (IFN-γ release levels, measured in International Unit per millilitre, IU/mL) in plasma was measured using CLIA (Liason, QuantiFERON -TB® Gold Plus Diasorin, Saluggia, Italy, cat no./ID #3110).

The QuantiFERON SARS-CoV-2-assay uses two mixes of SARS-CoV-2 spike glycoprotein-derived peptides (Ag1 and Ag2) selected to activate $CD4^+$ (Ag1) or $CD4^+$ and $CD8^+$ T-cells (Ag2), a negative control and positive phytohemagglutinin control[18]. According to the manufacturer's recommendations, an IFN-γ level $\geq 0.15$ IU/mL higher than that in the nil tube was interpreted as reactive (positive response).

The QFM is a one tube IGRA that assesses the combined activation of the adaptive immune response by stimulating T-cells using anti-CD3 antibodies (T-cell stimulant), and the innate immune response by stimulating T-cells with the Toll-like receptor 7/8 ligand R848. The results were divided into low response IFN-γ < 15 IU/mL (high risk for infections), normal response IFN-γ = 15–1000 IU/mL (low risk for infections), and high IFN-γ > 1000 IU/mL (low risk for infections – possible immune hyperreactivity)[19,20].

## Statistical analyses
Demographic data were reported according to vaccine type as percentages or medians with interquartile range (IQR). IQR was defined as the range between the 25th percentile (Q1) and the 75th percentile (Q3) of the data when sorted in ascending order.

The Mann–Whitney U test was used to compare antibody levels at both time measurements including the impact of vaccine type. The Kruskal-Wallis test was used to examine the overall differences in antibody and cellular response across different groups categorised by previous SARS-CoV-2 infection and/or a booster dose. Subsequently, pairwise comparisons were performed using Dunn's test, with adjustments made for multiple comparisons through Bonferroni correction.

Linear regression models were used to compare total S-Ab responses and SARS-CoV-2 specific T-cell responses in different groups and to determine factors influencing the immune response. We adjusted for the following covariates, identified a priori as potential confounders based on existing literature on vaccine-induced antibody responses[21–23]: age, gender, comorbidity (Charlson Comorbidity Index score), previous SARS-CoV-2 infection, booster vaccination, Inuit ethnicity, vaccine type and time interval (weeks) between the second vaccine dose and blood sampling.

Charlson Comorbidity Index scores[24] were calculated based on health data extracted from the Greenlandic Electronic Medical Patient Registry COSMIC. The score assigns a weight (1, 2, 3, or 6) to each of 19 major disease categories and is a validated measure of comorbidity[23]. We categorised the scores as 0, 1, and 2 or more.

Linear regression models were performed using log10-transformed antibody/IFN-γ release levels (outcome). A 95% confidence interval was estimated using linear regression models. All tests were two-sided and $p \leq 0.05$ were considered statistically significant.

Spearman's correlation test was used to evaluate the association between total S-Ab levels in saliva and plasma, as well as between total S-Ab levels in plasma and IFN-γ release levels.

Data analysis and visualisation were conducted using R, version 4.2.3[25].

## Ethics declaration
The study was approved by the Greenlandic Committee on Research Ethics (KVUG 2021-12) and the Greenlandic Health Authorities. The study was completely voluntary and rested on oral and written informed consent in either Greenlandic or Danish at enrolment. Participants were not paid.

## Reporting summary
Further information on research design is available in the Nature Portfolio Reporting Summary linked to this article.

## Results
A total of 430 individuals participated in the study (~1% of the Greenlandic population), with 64% being females and 68% Inuit at a median age of 46.5 years (IQR 33–57 years) for the entire cohort (Table 1). 409 (96%) were vaccinated with the mRNA-1273 vaccine (Moderna), while 15 (4%) received the BNT162b2 vaccine (Pfizer–BioNTech).

The 11-month follow-up after the second vaccine dose was completed by 323 (75%) individuals of whom 107 also provided a saliva sample and 78 provided an additional blood sample for measurement of T-cell responses. One-hundred and seven (25%) individuals were lost to follow-up due to unreachability. Among the 323 individuals who participated in the 11-month follow-up (97% vaccinated with the mRNA-1273/Moderna vaccine), 274 (85%) had received an additional booster dose homologous to the primary vaccine. The time between the second and third doses varied from 19 to 47 weeks (median 24 weeks) (Table 1).

### Total plasma antibody response
96% of participants had a protective level of total S-Abs two months post-primary vaccination (Time 1), while the remaining 4% of participants had insufficient (non-protective) levels of total S-Abs, signifying a hyporesponsive reaction to the vaccine (Fig. 1a).

11 months post-primary vaccination (Time 2), protective levels of total S-Abs were observed in 98% of participants, while 2% had non-protective levels (Fig. 1a). Over time, there was an increase in overall antibody levels (Supplementary Table 1). The median level of total S-Abs across all sampled time points reached 15942 kBAU/L (IQR 135–25000 kBAU/L) at Time 2, marking a fivefold rise compared to Time 1 ($p < 0.001$). At Time 2, 64% of

participants had reported having had a breakthrough infection, and an additional 6% reported having had a reinfection with a median time between positive tests of 4 months (IQR 3–4 months).

**Table 1 | Demographic characteristics of participants at study enrolment**

| Characteristics | Total (*N* = 430) | mRNA-1273 (Moderna) (*N* = 415) | BNT162b2 (Pfizer/ BioNTech) (*N* = 15) |
|---|---|---|---|
| Age at enrolment (years) | | | |
| Median (IQR) | 46.5 (33–57) | 46 (33–56) | 65 (54–70) |
| *N* Person (%) | | | |
| Male | 156 (36.3) | 151 (36.4) | 5 (33.3) |
| Female | 274 (63.7) | 264 (63.6) | 10 (66.7) |
| *N* Ethnicity (%) | | | |
| Inuit | 292 (68.0) | 278 (67.0) | 14 (93.3) |
| Mixed | 120 (27.9) | 120 (28.9) | 0 (0) |
| Non-Inuit | 18 (4.1) | 17 (4.1) | 1 (6.7) |
| *N* Residency (%) | | | |
| Nuuk | 312 (72.5) | 298 (71.8) | 14 (93.3) |
| Ilulissat | 113 (26.3) | 112 (27.0) | 1 (6.7) |
| Other | 5 (1.2) | 5 (1.2) | 0 (0) |
| Charlson Comorbidity Score Index (%) | | | |
| 0 | 353 (82.0) | 344 (82.9) | 9 (60.0) |
| 1 | 59 (13.7) | 55 (13.3) | 4 (26.6) |
| 2 or more | 18 (4.2) | 16 (3.9) | 2 (13.3) |
| Median (IQR) | | | |
| Interval (weeks) between dose 1 and dose 2 | 4 (4–5) | 4 (4–5) | 4 (4-4) |
| Interval (weeks) between dose 2 and dose 3 | 24 (23–24) | 24 (23–24) | 38.5 (34–39) |

Grouped by vaccine type (mRNA-1273 or BNT162b2).

Six participants (1.4%) had positive levels of total N-Abs at Time 1, indicating a low level of community transmission in the autumn of 2021[15]. At Time 2, 238 (74%) participants had become seropositive for total N-Abs. According to self-reported information, most cases were either asymptomatic or presented with mild symptoms. Only four participants (2%) experienced moderate symptoms but were not hospitalised. No COVID-19-related deaths were officially registered among the participants.

**Independent factors influencing the plasma antibody response**

Increasing time between primary vaccination and blood sampling and age had a negative effect on the total S-Ab response at Time 1 (Fig. 2 and Supplementary Data 1). Specifically, individuals above 60 years exhibited 30% (95% CI 19–39%) lower antibody levels compared to younger individuals (Figs. 1a and 2). Additionally, individuals with a Charlson Comorbidity Index score of 2 or more had 23% (95% CI 5–37%) lower antibody levels compared to healthier individuals. However, after adjusting for age, this association was no longer statistically significant (*p*-value 0.161) (Supplementary Data 1).

At Time 2, we also observed a 21% (95% CI 7–33%) reduced antibody response in individuals aged 60 years and above (Figs. 1a, 3 and Supplementary Data 1). No association between antibody response and underlying morbidity was found at Time 2 (*p*-value 0.382). Additionally, individuals who had received an early booster vaccination during the study period showed slightly lower antibody responses (1%, 95% CI 0–2%) compared to those who received a late booster dose. However, after adjusting for age, time since booster vaccination was not significantly associated with the antibody response (*p*-value 0.652) (Supplementary Data 1).

At Time 1, total S-Ab levels were 48% (95% CI 36–58%) lower in participants who had been vaccinated with the BNT162b2 (Pfizer/BioNTech) vaccine compared to those vaccinated with the mRNA-1273 (Moderna) vaccine (*p* < 0.001). However, at Time 2, no difference was noted between the two types of mRNA vaccines (*p*-value 0.571) (Figs. 2, 3, Supplementary Data 1 and Supplementary Fig. 2).

We found no difference in the total S-Ab response by Inuit ethnicity (Figs. 1b, 2 and 3 and Supplementary Data 1).

Previous SARS-CoV-2 infection (based on positive levels of total N-Abs) was positively associated with higher total S-Ab levels at both time points (*p* < 0.001) (Figs. 2 and 3 and Supplementary Data 1). The same effect was observed for booster vaccination, although not statistically significant (Supplementary Data 1). The combined effect of previous SARS-CoV-2

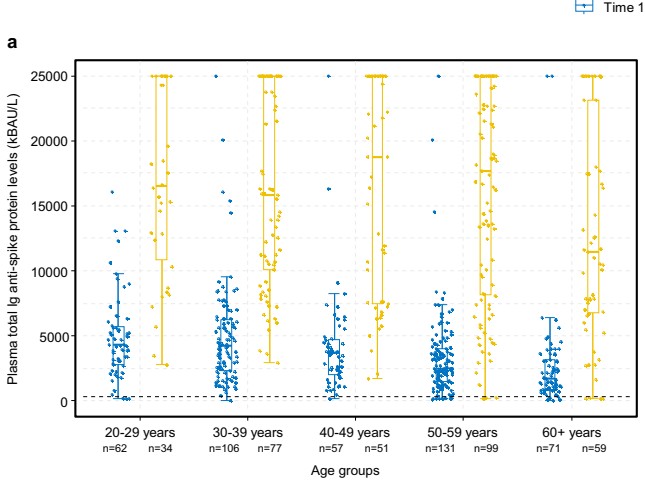
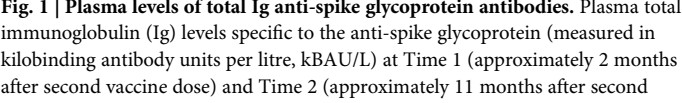
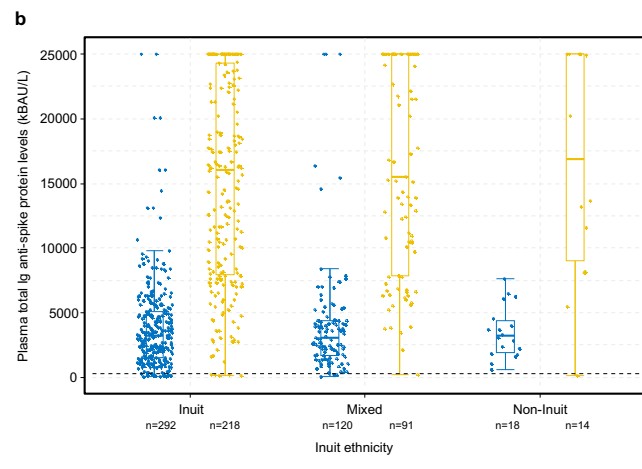

**Fig. 1 | Plasma levels of total Ig anti-spike glycoprotein antibodies.** Plasma total immunoglobulin (Ig) levels specific to the anti-spike glycoprotein (measured in kilobinding antibody units per litre, kBAU/L) at Time 1 (approximately 2 months after second vaccine dose) and Time 2 (approximately 11 months after second vaccine dose). Plasma levels by age groups (**a**) and Inuit ethnicity (**b**). The boxplots present the lower quartile, median, and upper quartile, and the error bars indicate 95% CI. *P* < 0.05 were considered significant. Dotted line indicates a cut-off level of 300 kBAU/L (protective level).

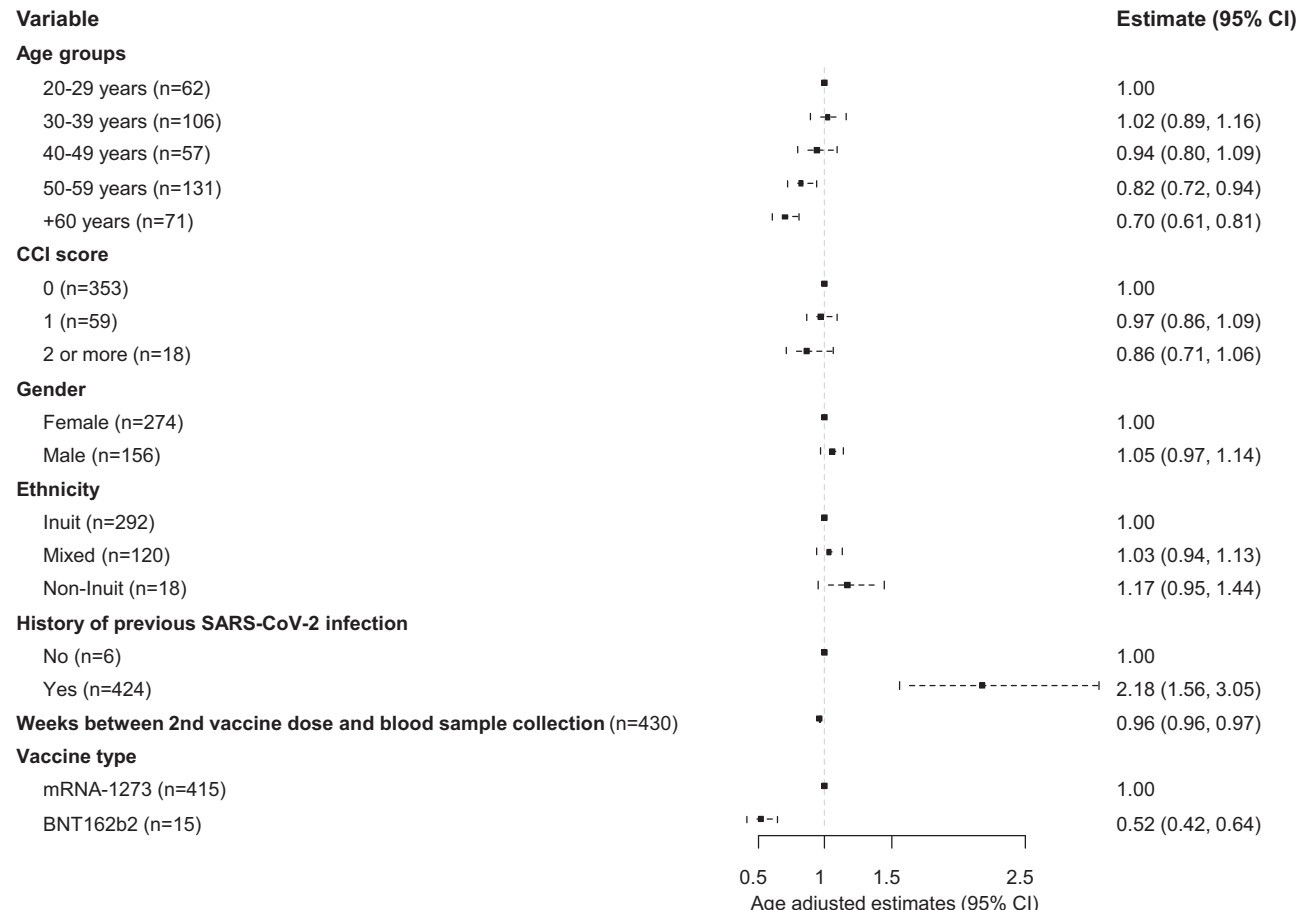

| Variable | | Estimate (95% CI) |
|---|---|---|
| **Age groups** | | |
| 20-29 years (n=62) | | 1.00 |
| 30-39 years (n=106) | | 1.02 (0.89, 1.16) |
| 40-49 years (n=57) | | 0.94 (0.80, 1.09) |
| 50-59 years (n=131) | | 0.82 (0.72, 0.94) |
| +60 years (n=71) | | 0.70 (0.61, 0.81) |
| **CCI score** | | |
| 0 (n=353) | | 1.00 |
| 1 (n=59) | | 0.97 (0.86, 1.09) |
| 2 or more (n=18) | | 0.86 (0.71, 1.06) |
| **Gender** | | |
| Female (n=274) | | 1.00 |
| Male (n=156) | | 1.05 (0.97, 1.14) |
| **Ethnicity** | | |
| Inuit (n=292) | | 1.00 |
| Mixed (n=120) | | 1.03 (0.94, 1.13) |
| Non-Inuit (n=18) | | 1.17 (0.95, 1.44) |
| **History of previous SARS-CoV-2 infection** | | |
| No (n=6) | | 1.00 |
| Yes (n=424) | | 2.18 (1.56, 3.05) |
| **Weeks between 2nd vaccine dose and blood sample collection** (n=430) | | 0.96 (0.96, 0.97) |
| **Vaccine type** | | |
| mRNA-1273 (n=415) | | 1.00 |
| BNT162b2 (n=15) | | 0.52 (0.42, 0.64) |

Age adjusted estimates (95% CI)

**Fig. 2 | Independent factors associated with the COVID-19 vaccine-induced antibody response 2 months after second vaccine dose.** Age-adjusted estimates and 95% CI from log10-linear regression models according to plasma total immunoglobulin (Ig) levels specific to the anti-spike glycoprotein (measured in kilobinding antibody units per litre, kBAU/L) at Time 1 (approximately 2 months after second vaccine dose). CCI score Charlson Comorbidity Index score.

infection and booster vaccination on the humoral and cellular response is illustrated in Fig. 4. Participants who had had a previous SARS-CoV-2 infection exhibited higher antibody levels, regardless of whether they had received a booster dose, compared to other groups ($p < 0.001$).

### Salivary antibody response
The median concentration of salivary total S-Abs from 107 participants at Time 2 was 7.34 kBAU/L (IQR 3.94–14.0 kBAU/L). Individuals with previous SARS-CoV-2 infection had significantly higher levels of salivary total S-Abs compared to individuals with no previous history of infection ($p < 0.001$) (Fig. 4b). Other factors such as gender, age, and Inuit ethnicity did not affect the salivary S-Ab response (Fig. 5 and Supplementary Table 2).

### T-cell immune-mediated response
General and specific SARS-CoV-2 spike glycoprotein T-cell responses were determined in 78 participants, 11 months after primary vaccination (Time 2). Eighteen participants (23%) had high levels of QFM IFN-γ, while 60 participants (77%) had moderate levels. The median concentration of QFM IFN-γ was 599 IU/ml (IQR 232–924). None of the participants had low levels of QFM IFN-γ. The SARS-CoV-2 specific CD4$^+$ + CD8$^+$ IFN-γ release showed that 47 participants (85%) had IFN-γ levels above the cut-off index and 8 participants (15%) did not. We did not find any association with gender, previous SARS-CoV-2 infection, or Inuit ethnicity (Figs. 4c and 6b). However, age above 60 years was associated with a reduced SARS-CoV-2 specific T-cell response, showing a 36% (95% CI 9–55%) decrease in IFN-γ release levels. (Fig. 6a and Supplementary Table 3).

### Correlation between humoral and cellular response
Levels of total salivary S-Abs positively correlated with plasma S-Ab levels as illustrated in Fig. 7a. On the contrary, we did not find a correlation between plasma S-Ab levels and SARS-CoV-2 specific CD4$^+$ or CD4$^+$ + CD8$^+$ IFN-γ release levels (Fig. 7b).

### Discussion
In this study, we present estimates of the effects on the humoral and cellular antibody response in a Greenlandic population vaccinated with two COVID-19 mRNA vaccines up to 11 months after the second vaccine dose.

Overall, we found a robust antibody response in the majority of the study population following vaccination with two doses of both the mRNA vaccines, with less than five percent exhibiting hyporesponsitivity. This equals what has been observed in other populations[23,26,27]. Indicating that the antibody response is shaped by a variety of influencing factors, we found that previous SARS-CoV-2 infection and booster vaccination had a significant additive effect on the antibody response, similar to findings in other populations[8,28–30]. This led to higher levels of total S-Abs up to 11 months after primary vaccination in our study population.

As in other studies, age above 60 years and underlying morbidity had negative impacts on the antibody response[9,23,29], although the association between underlying morbidity and antibody response was no longer significant after adjusting for age. This underlines the importance of giving priority to particularly the elderly, and individuals with comorbidities in the distribution of booster vaccines.

Notably, two months after the second vaccine dose, vaccination with the BNT162b2 (Pfizer/BioNTech) vaccine was associated with reduced

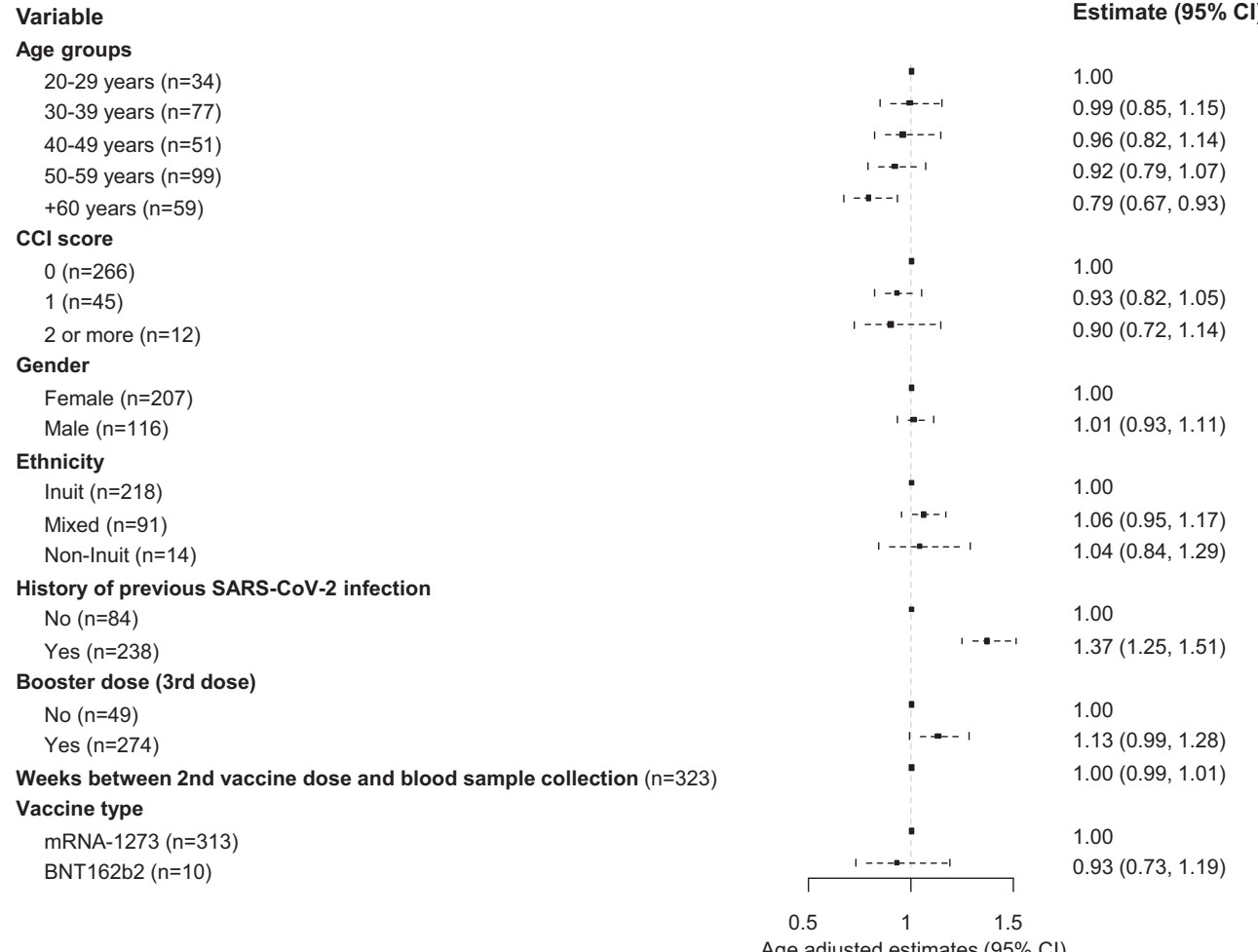

| Variable | | Estimate (95% CI) |
|---|---|---|
| **Age groups** | | |
| 20-29 years (n=34) | | 1.00 |
| 30-39 years (n=77) | | 0.99 (0.85, 1.15) |
| 40-49 years (n=51) | | 0.96 (0.82, 1.14) |
| 50-59 years (n=99) | | 0.92 (0.79, 1.07) |
| +60 years (n=59) | | 0.79 (0.67, 0.93) |
| **CCI score** | | |
| 0 (n=266) | | 1.00 |
| 1 (n=45) | | 0.93 (0.82, 1.05) |
| 2 or more (n=12) | | 0.90 (0.72, 1.14) |
| **Gender** | | |
| Female (n=207) | | 1.00 |
| Male (n=116) | | 1.01 (0.93, 1.11) |
| **Ethnicity** | | |
| Inuit (n=218) | | 1.00 |
| Mixed (n=91) | | 1.06 (0.95, 1.17) |
| Non-Inuit (n=14) | | 1.04 (0.84, 1.29) |
| **History of previous SARS-CoV-2 infection** | | |
| No (n=84) | | 1.00 |
| Yes (n=238) | | 1.37 (1.25, 1.51) |
| **Booster dose (3rd dose)** | | |
| No (n=49) | | 1.00 |
| Yes (n=274) | | 1.13 (0.99, 1.28) |
| **Weeks between 2nd vaccine dose and blood sample collection** (n=323) | | 1.00 (0.99, 1.01) |
| **Vaccine type** | | |
| mRNA-1273 (n=313) | | 1.00 |
| BNT162b2 (n=10) | | 0.93 (0.73, 1.19) |

0.5 1 1.5
Age adjusted estimates (95% CI)

**Fig. 3 | Independent factors associated with the COVID-19 vaccine-induced antibody response 11 months after second vaccine dose.** Age-adjusted estimates and 95% CI from log10-linear regression models according to plasma total immunoglobulin (Ig) levels specific to the anti-spike glycoprotein (measured in kilobinding antibody units per litre, kBAU/L) at Time 2 (approximately 11 months after second vaccine dose). CCI score Charlson Comorbidity Index score.

levels of plasma S-Abs. However, this finding should be interpreted with caution, as only 15 persons received the BNT162b2 (Pfizer/BioNTech) vaccine. Furthermore, this association was not observed at the 11-month follow-up.

One of the objectives of this study was to assess the SARS-CoV-2 vaccine-induced immune response in an Inuit population. Prior research indicates that ethnicity may impact vaccine response[10,22]. In addition, a specific genetic trait has been identified in a small proportion of Inuit that provides a specific susceptibility to respiratory tract infections, and it can be speculated whether genetic factors affect the response to vaccines[31]. However, in this study, which, to our knowledge, is the first study of immune response to SARS-CoV-2 vaccines in an Inuit population, we found no noticeable difference in the vaccine-induced immune response between Inuit and non-Inuit individuals.

Determination of both the T-cell and salivary S-Ab responses provides additional insights into the overall vaccine-induced immune responses. While the significance of oral mucosal immune responses in preventing respiratory infections is well-recognised[32], this is only beginning to be explored in the context of SARS-CoV-2 infection[33,34]. The presence of detectable levels of S-Abs in the saliva of all study participants may therefore serve as a contributing factor in reducing viral load leading to less severe disease progression. Our data suggest this possibility but do not offer a conclusive answer. We found a positive association between total S-Ab levels in plasma and saliva. This association could be explained by a spill-over of antibodies from serum into the saliva as previous research has found[35]. However, our findings suggest that oral mucosal secretions may serve as a potential predictive marker of protective immunity to SARS-CoV-2 infection following vaccination. Using a simple saliva test for antibody measurement may prove beneficial, particularly in settings where collecting blood samples may not always be feasible. Saliva-based SARS-CoV-2 antibody testing has been suggested as a substitute for serum-based testing[35], but notable individual-level variations observed in this study might raise concerns about the reliability of oral mucosal immunity testing.

Contrary to other findings[9], we did not observe a correlation between the specific SARS-CoV-2 spike T-cell response and the total S-Ab response in plasma. This discrepancy was not due to an insufficient general T-cell response (QFM assay) given that no participants had low levels of IFN-γ release. The lack of correlation could be due to the assays used and the peptides they use for T-cell activation; however, it is not entirely clear why our results differ from previous findings. Additionally, we observed an additive effect of prior natural infection on the salivary antibody response, although this effect was not reflected in the specific T-cell response. This was contrary to findings from other studies[9,36,37] with the exception in cases of severe disease where the T-cell response was impaired[38].

Although we found a robust immune response in our study population up to 11 months following primary vaccination, a substantial proportion of participants (74%) had experienced infection between our data collection points. Consistent with other studies[39,40], this suggests limited vaccine protection against infection with the Omicron variant, which was dominant in this period. However, it is noteworthy that despite many individuals being

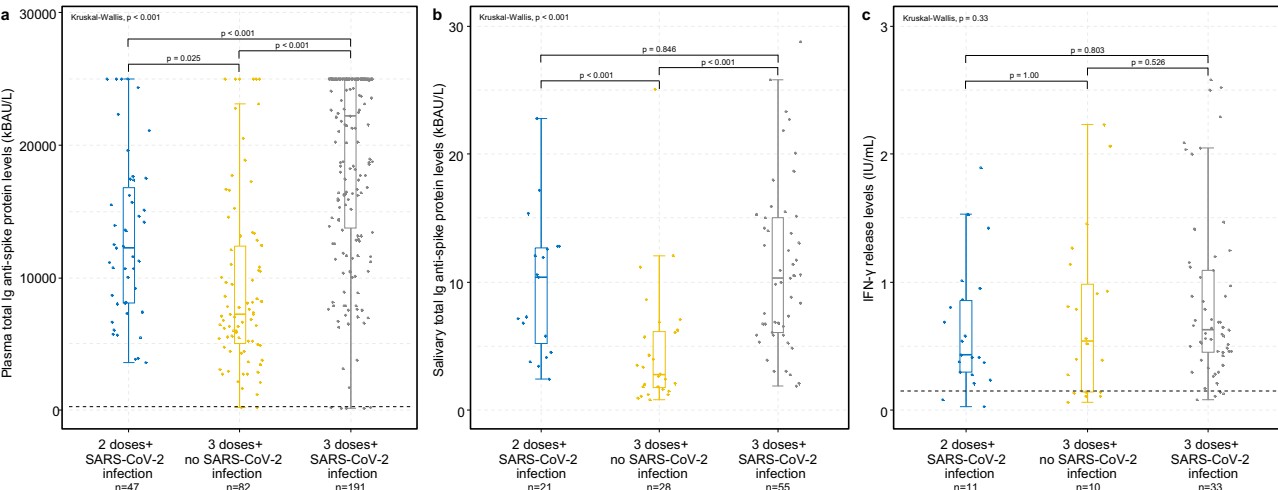

**Fig. 4 | Impact of previous SARS-CoV-2 infection and/or booster dose (third dose) on the COVID-19 vaccine-induced immune response.** Levels stratified by vaccination and infection status at Time 2 (approximately 11 months after second vaccine dose). **a** Plasma total immunoglobulin (Ig) levels specific to the anti-spike glycoprotein (measured in kilobinding antibody units per litre, kBAU/L). Dotted line indicates a cut-off level of 300 kBAU/L (protective level). **b** Saliva total immunoglobulin (Ig) levels specific to the anti-spike glycoprotein (measured in kilobinding antibody units per litre, kBAU/L). **c** SARS-CoV-2-specific T-cells (CD4$^+$/CD4$^+$ + CD8$^+$) IFN-γ release plasma levels (measured in International Unit per millilitre, IU/mL). Dotted line indicates a cut-off level of 0.15 IU/mL (positive reaction). The boxplots present the lower quartile, median, and upper quartile, and the error bars indicate 95% CI. Kruskal-Wallis test was used to examine the overall differences in antibody and cellular response across different. Pairwise comparisons were performed using Dunn's test, with adjustments made for multiple comparisons through Bonferroni correction. $P < 0.05$ were considered significant.

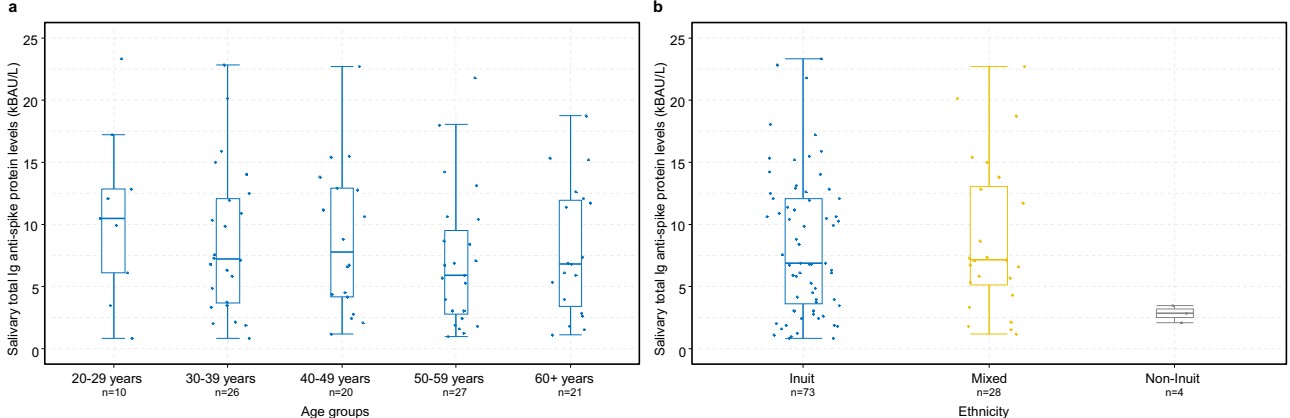

**Fig. 5 | Saliva levels of total Ig anti-spike glycoprotein antibodies.** Saliva total immunoglobulin (Ig) levels specific to the anti-spike glycoprotein (measured in kilobinding antibody units per litre, kBAU/L) at Time 2 (approximately 11 months after second vaccine dose). By age groups (**a**) and by Inuit ethnicity (**b**). The boxplots present the lower quartile, median, and upper quartile, and the error bars indicate 95% CI.

infected after vaccination, none reported serious illnesses or required hospitalisation. These observations further support the long-term protectiveness of the vaccines against severe disease and death[2,26,27]. This protection can likely be attributed to various immunological components, including the presence of oral mucosal antibodies and the SARS-CoV-2 specific T-cells. ls, it is evident from other studies that the T-cell responses play an important role in the long-term defence against COVID-19[41,42]. A recent study emphasised the vital role of cellular immunity in vaccine protection against severe SARS-CoV-2 disease, which could be particularly relevant in the case of Omicron, known for its ability to evade antibody responses[42]. Notably, even in unvaccinated individuals with no previous infection, SARS-CoV-2 reactive T-cells have been detected, which may indicate cross-reactivity with other human coronaviruses[38,43]. However, it remains uncertain whether these T-cells can contribute to clinical protection against COVID-19[38,44].

During the early stages of the pandemic the Greenlandic Health Authorities, like authorities in other geographically isolated countries, promptly implemented various travel restrictions to minimise and delay the introduction of the coronavirus into the country, allowing them more time to vaccinate the population, i.e., a containment strategy[45,46]. Based on historical evidence and as previously mentioned, the public health profile, the population was supposed to be at a higher risk of severe illness and mortality from SARS-CoV-2 infection, which made the vaccination efforts even more critical[6].

The combined strategy proved successful as COVID-19 only made a substantial impact in terms of community transmission in Greenland in the autumn of 2021 (Supplementary Fig. 1) when most of the population had received the standard two-dose regimen of one of the two mRNA vaccines, providing presumed good protection against the virus. While Greenland had a relatively high cumulative COVID-19 incidence (21,419/100,000 inhabitants as of June 2022) not very different from countries like Denmark (57,798/100,000 inhabitants as of June 2022), the country maintained throughout the pandemic a very low case-fatality rate of 0.18% compared with a case-fatality rate of 0.30% in Denmark[3,7]. This indicates an effective

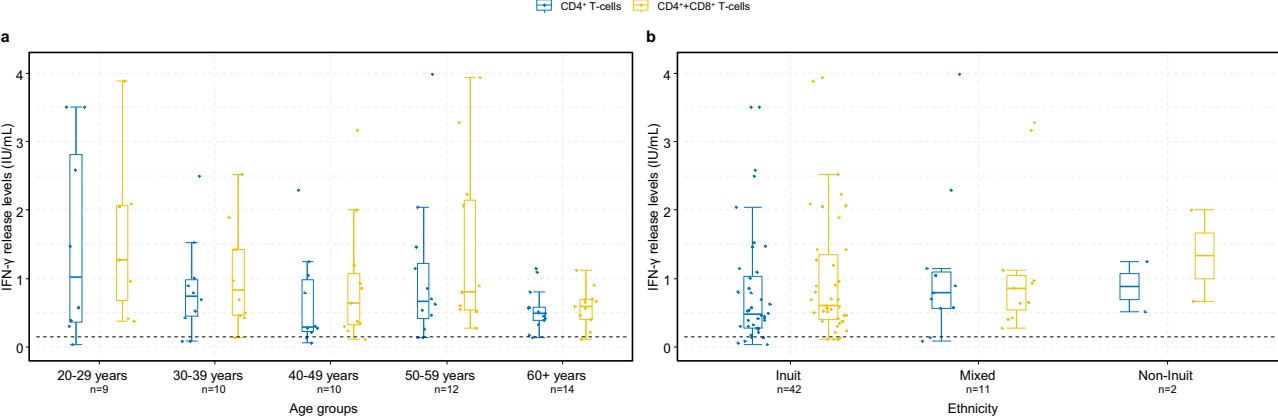

**Fig. 6 | SARS-CoV-2-specific T-cell (CD4$^+$/CD4$^+$ + CD8$^+$) IFN-γ release plasma levels.** T-cell interferon-γ release specific to the SARS-CoV-2 spike glycoprotein (measured in International Unit per millilitre, IU/mL) at Time 2 (approximately 11 months after second vaccine dose). By age groups (**a**) and by Inuit ethnicity (**b**).

The boxplots present the lower quartile, median, and upper quartile, and the error bars indicate 95% CI. Dotted line indicates a cut-off level of 0.15 IU/mL (positive reaction).

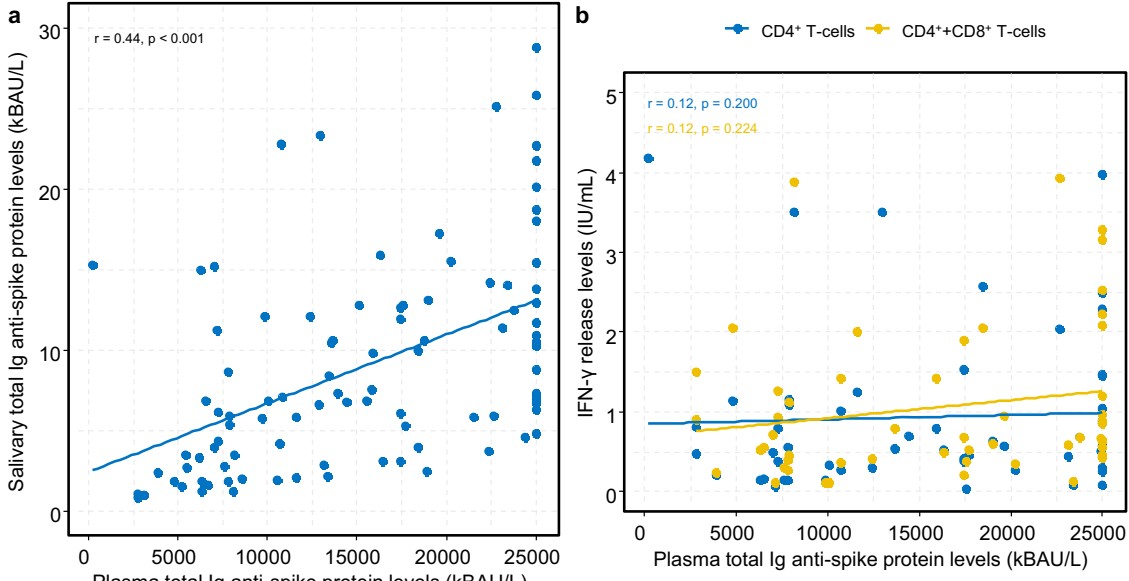

**Fig. 7 | Correlations in immune responses. a** Shows the correlation between total immunoglobulin (Ig) specific to the anti-spike glycoprotein (measured in kilobinding antibody units per litre, kBAU/L) in plasma and saliva approximately 11 months after the second vaccine dose (*n* = 107 individuals). **b** Illustrates the correlation between plasma levels of total immunoglobulin (Ig) specific to the anti-spike glycoprotein (measured in kilobinding antibody units per litre, kBAU/L) and SARS-CoV-2 specific T-cells (CD4$^+$/CD4$^+$ + CD8$^+$) IFN-γ release plasma levels (measured in International Unit per millilitre, IU/mL) approximately 11 months after the second vaccine dose (*n* = 55 individuals). Spearman rank correlation (r) was performed (two-sided). *P* < 0.05 were considered significant.

pandemic strategy that not only curtailed the spread of the virus but also shifted the peak of transmission to a period when milder SARS-CoV-2 variants (Omicron) were predominant in the country. The combined influence of delayed virus introduction, documented effective vaccines, and a robust general and specific immune response in the Greenland population as seen in this study, collectively contributed to a milder course of the pandemic in the country and largely spared Greenland the severe consequences of the pandemic.

The present data from Greenland contribute to an important and unexplored aspect of the pandemic's course in the population. As previously highlighted, there is limited research on the immunogenicity of COVID-19 vaccines in Inuit populations, who possess a unique environmental and possibly immunological profile and are at increased risk of severe disease following COVID-19 infection[3,6]. Thus, findings from non-indigenous populations may not be generalisable to indigenous populations[11].

Therefore, the discovery that the Greenlandic population exhibits an immunological response to COVID-19 vaccination comparable to other populations is important and positive. Our findings strengthen the validity of the vaccine strategy implemented by the Greenlandic authorities, not only at present but also for future endeavours in terms of booster vaccination planning. Additionally, they could reduce booster hesitancy in the population. Our results also emphasise the potential use of similar models for future pandemic preparedness and response in other island countries and isolated communities.

Our study has some limitations. First, the sample size of our study is relatively small. However, compared with figures from The Population Survey in Greenland 2018[5] our study population is very representative of the Greenlandic population. Additionally, due to limited resources, we were only able to measure the antibody response at two time points over a 11-month period. This limited sampling frequency prevented us from fully

capturing the entire course of the antibody response. It would have been valuable to examine the antibody levels prior to primary vaccination and after a booster dose, as we know that booster doses can have a positive impact on antibody levels[8,47,48]. Furthermore, salivary antibodies and T-cell response measurements were only performed once at the 11-month follow-up. It would have been informative to track these markers throughout the entire vaccination process and at different time points to better understand their dynamics. Also, the assay used to analyse the total S-Ab response had an upper measurement limit of >25,000 kBAU/L, a threshold surpassed by numerous participants. Consequently, we could not fully depict antibody dynamics at very high levels. Moreover, data on the administration of booster doses and SARS-CoV-2 test results relied on self-reported information, which could lead to recall bias. Finally, we could only compare our findings among Inuit and non-Inuit in Greenland and not directly with other populations, e.g., western or other Arctic populations.

In conclusion, the population of Greenland exhibits a robust and enduring vaccine-immune response, both humoral and cellular, comparable to other population groups, approximately one year after receiving their second SARS-CoV-2 vaccination dose with either mRNA-vaccine. S-Ab responses declined with increasing age, consistent with findings from other populations, indicating that age should be taken into consideration in future vaccination campaigns. Booster vaccination and in particular previous SARS-CoV-2 infection positively affected S-Ab responses, resulting in high levels of antibodies in both plasma and saliva. Our findings have provided important insights into a crucial aspect of protection against COVID-19 in the Greenlandic population and add to the understanding of the mild course of the COVID-19 pandemic in Greenland. Additionally, they provide valuable insights for strategic pandemic preparedness in Greenland, as well as for other isolated populations or comparable geographical regions.

## Data availability

Limitations apply to data accessibility. Due to Greenlandic law and ethical considerations, the full dataset is not publicly available because it contains highly detailed and individually linked data. Unidentifiable source data is available in Supplementary Data files 2-7. To access raw data, researchers are required to obtain ethics approval from the Greenland Research Ethics Committee and consent from study participants. Contact the corresponding author for further details on these limitations and the specific conditions for accessing raw data.

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

## Acknowledgements

The authors of this study would like to thank The Board of Health, Greenland, Uka Wilhjelm Geisler, Head of the Department of Internal Medicine, Queen Ingrid's Hospital, Nuuk, Greenland, Inge-Lise Kleist, Head of Laboratory, Queen Ingrid's Hospital, and Ilulissat Regional Hospital, Ilulissat, Greenland, for giving access to technical facilities and staff assistance. We would also like to thank Nuuk ugeavis, Air Greenland, and Nuuk Center for their assistance in recruiting participants for the study. Finally, we would like to thank all the study participants. This research was funded by the Greenland Research Council and the Greenland Institute of Natural Resources PhD-grant (grant no. 80.39), Kong Christian den Tiendes Fond (grant no. 35/2022), Grosserer L.F.H. Foghts Fond (grant no. 22.288), and contributions from the Department of Clinical Microbiology, and the Department of Infectious Diseases, Rigshospitalet University Hospital, Copenhagen, Denmark, and the Department of Clinical Biochemistry, Bispebjerg Hospital, Copenhagen, Denmark.

## Author contributions

M.M. and A.K. conceived of the study. M.M., A.K., L.F.H., P.V., and K.M. designed the study and developed the study protocol. M.M. collected the samples. C.D.H., L.F.H., and N.K. carried out all laboratory analyses. M.M. wrote the first draft of the manuscript and performed the data analysis and visualisation with the assistance of M.A. All authors contributed to the interpretation of data and provided critical feedback on the paper. All authors had full access to the data and accept responsibility for submission. All authors contributed to the article and approved the submitted version.

## Competing interests

The authors declare no competing interests.
