## [Peer Review File · Communications Medicine]

Reviewers' comments:

Reviewer #1 (Remarks to the Author):

The manuscript elaborated by Møller et al. describes the evaluation of antibody levels against Spike protein in serum and saliva and cellular responses through the readout of IFN- γ release from T-cells after COVID-19 vaccination in Greenland up to 12 months after the administration of the first vaccine dose. Moreover, an evaluation of influencing factors on the immune responses is performed. However, some aspects of the study appear confusing and should be addressed.

1. In the abstract, the study aim defines the evaluation of the antibody levels at 12 months after vaccination in participants receiving a complete two-dose vaccination. However, in the results section is stated that 81% of the cohort had received an additional booster by Time 2. The study design description should be rephrased for an easier understanding of the study.

2. The aims of the study presented in the introduction should be rephrased since it appears confusing the fluid used to measure the T-cell responses (line 86).

3. Since part of the COVID-19 tests and administration of the booster dose information are collected based on self-reporting, this aspect should be highlighted as a study limitation.

4. It is confusing the time variable since it can be found within the text defined as 3 or 12 months from the first vaccine dose (lines 131-132) but it does not match the time variable in other parts of the manuscript (Supplementary Figure 1 – which is inconsistent with the timeline).

5. Include when the different vaccine doses were administered in Figure 1 to provide a better overview of the study design.

6. Mucosa immunity encompasses different fluids, since only oral mucosa as saliva collection is evaluated, the definition of mucosal antibodies/immunity should be redefined to oral mucosa or salivary antibodies.

7. In line 203 is stated the time range between the second and the third dose is 19 to 47 weeks (24 weeks median). According to Table 1, why did the individuals who were vaccinated with the BNT162b2 vaccine receive the booster so late? Moreover, could this influence the levels at Time 2? What was the vaccine type administered as a booster dose?

8. P-values are not indicated in figures, this complicates the reading flow of the results. In general, the quality of the figures should be improved. Figure 2 shows different colors for observed data that are not defined. The x-axis of Figure 3 appears as a continuous variable while should be categorical. Figure 4 observed data for the group "2 doses + no SARS-Cov-2 infection" are missing. Saliva antibodies and IFN- γ levels should be reported as done for serum antibodies (classified according to the infection groups, and doses... and not shown as a histogram – Figure 5 – also lacking definitions). Supplementary Figure 1 should be corrected.

9. Since age is an influencing factor in immune response upon vaccination and might be related to comorbidities, I suggest evaluating association using linear regressions adjusting for age.

10. State the non-significant p-values in Table 2 to provide a general overview for tendencies.

11. In which figure are presented results in lines 230-234? Are individuals with two and three doses combined when infection status is evaluated?

12. Has been used the same laboratory assay to quantify antibodies in both serum and saliva? Is this assay validated for both fluids with the same cut-off?

13. What is the correlation coefficient from the Spearman test between serum and saliva antibodies?

14. Which test and in which Figure/Table can be found the association between IFN- γ and the other factors? (Lines 263-264). In the method section, it is mentioned the evaluation of IFN- γ levels released from CD3+ cells, are these results used to calculate the specific CD4+/8+ IFN- γ levels, or how is the analysis performed??

15. Have the authors considered the possible spill-over of antibodies from serum into the saliva, indicating the association of these two fluids (lines 291-294)?

16. Have the authors considered the possible lack of correlation between IFN- γ and antibodies due to saturation of the serum assay? And the T-cell detection in non-infected individuals due to vaccination? (lines 314-316)

Minor comments:

17. Provide a reference relating to a) the vaccine roll-out or their implication in disease severity/death decrease (Line 56); b) regarding Greenland population (lines 96, 104, and 105); and c) regarding identification of potential cofounders (lines 179-180).

18. The first sentence appears a bit out of context since containment restriction lifts were implemented since transmission, hospital pressure, and disease severity decreased (line 57).

19. How many COVID-19 cases (lines 64-66)?

20. What were the criteria chosen by the authors for the reinfection definition? Line 206, what was the time between positive tests that 5.7% of participants with more than one positive test (reinfection)?

21. Correct the statistical test used in line 171 since the data analyzed is not paired.

22. Specify the antigen in line 228.

Reviewer #2 (Remarks to the Author):

Brief summary of the manuscript: Robust immune response to COVID-19 vaccination in Greenlanders – an observational study in an Island population
Møller et al. have analyzed the presence of antibodies against COVID-19 in the Greenland population during the following vaccination between the winters of 2021 and 2022.

Overall impression of the work:

The text would benefit from greater structural organization as well as adding in more contextual information regarding the purpose of the study. Below I recommend some writing improvements for clarity in some sections. The rationale of the study design requires further explanation specifically within the introduction section, we see very little information about the COVID-19 epidemiological scenario in Greenland. Given one of the main points of the study is to discuss the unusual transmission pattern observed in Greenland, the lack of specific numbers (e.g., number of cases) compromises contextualization and would be useful in demonstrating the overall impact of the study. The authors provide a sufficient description of statistical methods used to calculate any differences observed between the sample sizes. The sample size is appropriate for the questions being asked. Please, revise the reference list, some of them do not seem to provide enough information or do not follow any official citation styles (e.g., references number 3, 4).

Specific comments, with recommendations for addressing each comment:

Abstract:

- 1)Line 29: Include citation, Noahsen et al. Referenced in the introduction line 60
- 2)Line 35: Include the number of participants included in the T-cell analyses
- 3)Line 39-41: Include numerical data to demonstrate booster doses and infection enhanced antibody response. Additionally, include data to support the conclusion that comorbidities

weakened antibody response.

4)Line 47: Include citation(s) for other "population groups"

Introduction:

1)Lines 64-66: The following sentences should read: "Despite good vaccination coverage compared to other Arctic regions³, over the course of the pandemic, Greenland experienced a considerable number of COVID-19 cases. However, although few cases were seen prior to the advent of the Omicron (B.1.1.529) ..."

2)Lines 64-68: It would be interesting to explore further the COVID numbers for Greenland. "Considerable number," "Few cases," and "markedly increased" are too vague.

3)Line 71: Remove the word "possible"

4)Line 72: With the phrasing "relatively low" there should be a citation including the reference group, meaning include a citation with hospitalization and death rates for population(s) in June of 2022

5)Line 73: Include a citation for the number of hospitalizations

6)Line 77: Are there any previous studies that demonstrate that Greenlandic or other Inuit populations have different immune responses to vaccines? If so, I would include data from them to emphasize the importance of this study.

7)Line 81: Please give further explanation of what "limited healthcare services" entails, specifically? Additional background information would be beneficial to clarify how Greenland's population would be impacted differently than other countries would be helpful to strengthen the introduction.

8)Line 85: Sentence should read "Specific aims were to quantify the COVID-19 vaccine-induced antibody immune response, to analyze the SARS-CoV-2 specific total Ig response in serum, the T-cell response, and total Ig response in mucosa."

Methods:

1)Line 94-99: How does the fact that most of Greenland's population is of Inuit descent impact how the pandemic unfolded there? Is that a critical point? If yes, please discuss it further or remove it from the text.

2)Line 94-99: As discussed in the comments regarding the introduction, including some further details about the healthcare system in Greenland would help contextualize the paper.

3)Line 108-114: I would recommend moving this portion to the supplemental section of the article, it interrupts descriptions of participant demographics. If not, I would create a new section within the methods entitled "Participant enrollment."

4)Line 121: replace "enrolment" with enrollment

5)Lines 123 – 124: Please, define if breakthrough infection is infection after 1 dose, 2 doses, or 2 doses plus booster.

6)Line 128: Although not in the national system, self-reporting vaccination status could be proved with a personal vaccination card or similar? Please, describe.

7)Line 136: Replace "Salivette Cotisol" with Salivette Cortisol

8)Line 136: Include the company and cat. #

9)Line 140: This statement contradicts the statement made in Line 132, which states all participants had blood drawn 12 months after their first dose. In line 140 however, you say "another subset of participants had additional blood samples drawn at 12 months after the first vaccine." I would suggest clarifying the participants in each experimental group.

10)Line 142: Rearrange the sentence to say, "All samples were shipped frozen (-20 oC) to Denmark within four hours after processing for final laboratory analysis."

11)Line 143: Include the company and cat. #

12)Line 146: Replace the word "after" with "of"

13)Line 152: Explain how the cut-off index is calculated.

14)Line 155-156: Can you explain further the difference between "insufficient (non-protective)" and "not considered sufficient to neutralize."

15)Line 160: Include a cat. #

16)Line 162: The end of the sentence is missing a period after control

17)Line 167: One or two tailed?

18)Line 177: This is the first time that the impact of ethnicity is mentioned in the text. If that

should be discussed, it should be better explored in the introduction section.

19)Line 180: Include a citation to provide basis for the statement "based on existing literature"

Results:

1)Line 195: I would advise re-writing the section "study population and descriptive statistics" for clarity. The current paragraphs are confusing.

2)Line 198: Include that it was a 12-month follow-up after their second vaccine dose.

3)Line 198: Include the samples who gave saliva samples, are these the same 323 individuals who gave blood samples 12-months after their first vaccine.

4)Line 204: What was the distribution of Moderna and Pfizer vaccines of the people who returned for their 12-month follow-up? What booster did they receive?

5)Line 208: replace "covid-19" with COVID-19

6)Lines 212-213: Combine these two sentences for clarity: "At Time 1, 95.6% of participants demonstrated a significant level of total Ig anti-spike antibodies reaching peak levels at 4-6 weeks after the second vaccine dose, with a median of 3147 kBAU/L (IQR 32-25000 kBAU/L)" (Figure 2).

7)Line 216: Include a citation indicating that low levels of N-abs indicate low transmission levels (Whitaker et al.)

8)Line 217-221: Reword this paragraph for clarity, you first say that antibody levels were stable and then you say that median at Time 2 was five times greater than at Time 1.

9)Line 218-219: Is that the median for all timepoints? Are you comparing the two peak Ig anti-spike levels from Time 1 and 2?

10)Line 220-221: Indicate if these patients had covid (self-reported or officially notified).

11)Line 225: This is the first time the Charlson comorbidity index is mentioned in the text (besides being added to the epidemiological characterization table). Please justify why this is important and which comorbidities are being considered, include this information within the methods section

12)Line 227: Please include data to justify this claim.

13)Lines 240 – 246: The data presented here is not relevant and is not even mentioned in the discussion section.

14)Lines 257-261: Is measuring IFN- γ enough to be classified as a T-cell mediated immune response? The IFN- γ is an important aspect of the cellular response, however other parameters should be included to call it a T-cell mediated response.

Discussion:

1)Line 267: We cannot say that the experimental strategy and results presented in this paper are enough to call it an overview of the cellular response.

2)Lines 272-273: All results were presented in months after the first dose, but here you are referencing 4-6 weeks after the second dose. I believe that these timepoints should be clarified or edited within the figures, methods, and discussion sections for maximum clarity.

3)Line 279: The whole purpose of the study is only mentioned clearly in this section. Although the authors do discuss the importance of studying this specific population, this should be hinted at in the introduction more clearly. Specifically, the sentence on Line 280, beginning with, "In addition, a specific genetic trait" must be included within the introduction as basis for conducting the study and understanding the study's importance. The more details you can include regarding this, the stronger the paper.

4)Line 304: You include two claims that are contrary to previous literature, do you have any hypotheses for why this may be? If so, include reasonings or citations for these beliefs.

5)Line 307: Were most of these infections between doses or shortly after receiving their second dose? I would caution against claiming that the vaccination series has limited protection against infection.

6)Line 316: Do you have a citation to justify that cross-reactivity may be responsible for the T-cell response?

7)Line 318-342: Is this section being published along with the paper? If not, discussing travel restrictions and border closure is a crucial point to explain the numbers registered in Greenland and must be mentioned in detail in the introduction and/or discussion sections.

Figures and tables:

1)Table 1: I would recommend table one to have only the main demographic information (age,

- sex, vaccine type, presence of co-morbidities). Some aspects are not discussed in the paper such as smoking and alcohol habits. Data should be transferred to a supplementary table or removed.
- 2)Table 2: Include on the statistically significant p-values for all rows in the table
 - 3)Figure 1: I think this could be moved to the supplementary material, as it does not demonstrate any "new" data that you have produced.
 - 4)Figure 2: As mentioned above, switching between months after the first dose and weeks after the second dose is confusing. I would suggest potentially changing the axes titles for greater clarity.
 - 5)Figure 2: Why are there colored dots (green for Time 1 and red for Time 2) and gray dots? Do they demonstrate different participants? The green and red dots also are larger than the gray dots, is this purposeful? If so, a legend is required to distinguish these differences.
 - 6)Figure 3: See comments above regarding the Time 1 and Time 2 and axes labels using weeks since second dose.
 - 7)Figure 3: Why is the box for Time 1 Inuit participants not encapsulating most of the data points? Additionally, for why is the box between weeks 20-40 whereas Mixed and Non-Inuit populations have most data points between 0-20 weeks.
 - 8)Figure 4: Bars in this figure are too big, I would suggest adjusting for a better visual impression.
 - 9)Figure 4: There are no points, just a line for the first column titled "2 doses + no SARS-CoV-2 infection", was there only one participant? If so, I would remove this column from the comparison.
 - 10)Figure 5: I would adjust how this figure is formatted. It is a bit complicated; I would begin by including the spearman correlation values on the graphs. I would also include the bar graphs on one row and the correlation graphs on another with axes titles.
 - 11)Supplementary Figure 1: I still believe that the axes would be clearer if they were marked regarding weeks/months since the first dose. However, if you would like to keep them labelled as is, I would recommend including this figure in the main text.
 - 12)Supplementary Figure 2: Include number of participants in each cohort in the methods section.
 - 13)Overall: Just as a general note, I would recommend changing the colors for this figure. People who are colorblind are likely unable to distinguish between red and green, I would suggest using something like blue and yellow.
 - 14)Overall, for all figures where comparisons are made, please include either asterisks or p-values to visualize the significant comparisons. Also include details of the statistical analyses done for each figure.
 - 15)Overall: Include (n=) values within your figures to help provide clarity on how many people are included in each analysis.

Reviewer #3 (Remarks to the Author):

In this study Møller et al provide a in interesting overview of humoral and cellular immunity in Greenland populations. Given the peculiarity of Greenland geography and ethnic makeup this study is of great interest. I have a few suggestions that could significantly improve the manuscript.

Major:

- The authors discuss the importance of the analysis in an Inuit populations and the reviewer agrees this is most novel and important aspect of the manuscript. The authors should try to expand this point by showing:
 - ♣ A direct comparison between Inuit and non-Inuit
 - ♣ Data on mucosal antibodies
 - ♣ Data on cellular responses
- Data that is discussed in the results is not always shown in figures, this makes the data more difficult to interpret. Specifically, lines 249-254 and lines 258-261. Figure 5 probably contains some of this data (correlation between serum and mucosal IgG) but it is really hard to read. Axes should be labeled, and plots separated for better understanding, the p values and R values should be placed inside the plot to aid interpretation. The significance of the bar plots is unclear to me.
- The most relevant data presented in table 2 and sup table 1 should be shown in form of a figure (at least the data mentioned in the results) so it is easier to understand. Especially the data on the relationship with age and antibody levels would be interesting to see, have the authors done a similar analysis for the cellular response?

Minor:

- Quite obviously it is impossible to judge humoral immunity 12 months after vaccination when 81% of the individuals received a booster and at least 65% seem to have had a breakthrough infection. Infection especially greatly influences the antibody levels at Time2 (Figure 4). This is noted in the discussion, but the authors should make this point much clearer throughout the manuscript and mention this whenever they draw conclusions that antibody levels at 12 months are “stable” for example line 218. The authors could also show the relationship between the time since booster/infection and antibody levels, either with a correlation of by binning patients into early and late.
- The 3 month and the 12 month time points actually comprise a fairly wide range of timepoints (figure 2) this should be stated more clearly in the text when the study design is presented
- Line 217-218: “At Time 2, 2.2% of participants had non-significant levels of total Ig anti-spike antibodies, while 97.8% had significant levels.” Please clarify significant compared to what, baseline?

Point-by-point response to referees

Reviewer #1 (Remarks to the Author):

The manuscript elaborated by Møller et al. describes the evaluation of antibody levels against Spike protein in serum and saliva and cellular responses through the readout of IFN- γ release from T-cells after COVID-19 vaccination in Greenland up to 12 months after the administration of the first vaccine dose. Moreover, an evaluation of influencing factors on the immune responses is performed. However, some aspects of the study appear confusing and should be addressed.

We thank reviewer 1 for their careful consideration of our manuscript and appreciate the opportunity to reflect on and respond to their comments.

1. In the abstract, the study aim defines the evaluation of the antibody levels at 12 months after vaccination in participants receiving a complete two-dose vaccination. However, in the results section is stated that 81% of the cohort had received an additional booster by Time 2. The study design description should be rephrased for an easier understanding of the study.

We have rephrased the method section in the abstract for clarity

2. The aims of the study presented in the introduction should be rephrased since it appears confusing the fluid used to measure the T-cell responses (line 86).

We have rephrased the sentence for clarity (line 98-100).

3. Since part of the COVID-19 tests and administration of the booster dose information are collected based on self-reporting, this aspect should be highlighted as a study limitation.

Very good point which have been added to the discussion section on study limitations (line 387-388).

4. It is confusing the time variable since it can be found within the text defined as 3 or 12 months from the first vaccine dose (lines 131-132) but it does not match the time variable in other parts of the manuscript (Supplementary Figure 1 – which is inconsistent with the timeline).

We agree that the study design, and more specifically the time variable, requires a better definition.

Originally, we designed the study with inspiration from a similar Danish study with which we aimed to collaborate, hoping to include additional sample collection time points, i.e., between the first and second doses. However, due to a lack of funding, this was not possible. Consequently, the strategy and design of the study, particularly concerning sample collection time points, underwent changes during the study timeline. Given the current situation, we have decided to present the results in terms of time since the second vaccine, as this aligns with our inclusion criteria and provides a better overview of our design and results.

Therefore, we have corrected the definition of the time variable in the text and all relevant tables and figures.

5. Include when the different vaccine doses were administered in Figure 1 to provide a better overview of the study design.

We have revised and combined Figure 1 and Supplementary Figure 1 do provide a better and clearer overview of the study design. Please see the new Supplementary Figure 1, which also includes when the vaccine doses were administered.

6. Mucosa immunity encompasses different fluids, since only oral mucosa as saliva collection is evaluated, the definition of mucosal antibodies/immunity should be redefined to oral mucosa or salivary antibodies.

All sentences including the mention of mucosal antibodies/immunity have been changed to either oral mucosal immune response or salivary antibodies for a clearer definition.

7. In line 203 is stated the time range between the second and the third dose is 19 to 47 weeks (24 weeks median). According to Table 1, why did the individuals who were vaccinated with the BNT162b2 vaccine receive the booster so late? Moreover, could this influence the levels at Time 2? What was the vaccine type administered as a booster dose?

A possible explanation for the delayed administration of booster doses to individuals vaccinated with the BNT162b2 vaccine is related to logistical and resource conditions concerning vaccine distribution in Greenland early in the pandemic. Initially, when the BNT162b2 vaccine became available, Greenland prioritized vaccinating the older and vulnerable population, along with essential personnel such as healthcare workers in major towns like Nuuk. Subsequently, as the mRNA-1273 vaccine became an option, plans were made for vaccination in more remote parts of Greenland, including smaller towns and settlements. It was not until the spring/summer of 2021, when most of the adverse weather and ice conditions had subsided, that the Greenlandic health authorities could efficiently sail along the coastline with a large ship containing vaccines and healthcare personnel to vaccinate the population. Therefore, the Greenlandic health authorities did not have the capacity to initiate booster vaccinations for the entire population, regardless of the vaccine type, until the fall/winter of 2021.

As a result, individuals who had received the BNT162b2 vaccine early in 2021 received their booster doses much later than those more recently vaccinated with the mRNA-1273 vaccine. While we cannot entirely rule out that this delay may have influenced antibody levels at Time 2 to some extent, we do not believe it significantly affects the main results of the study (based on a linear regression model adjusted for time since booster dose, see Supplementary Table 1).

All participants who received a booster dose was vaccinated with the same vaccine type as the previous two doses. This has been added to line 225 for better clarity.

8. P-values are not indicated in figures, this complicates the reading flow of the results. In general, the quality of the figures should be improved. Figure 2 shows different colors for observed data that are not defined. The x-axis of Figure 3 appears as a continuous variable while should be categorical. Figure 4 observed data for the group “2 doses + no SARS-Cov-2 infection” are missing. Saliva antibodies and IFN- γ levels should be reported as done for serum antibodies (classified according to the infection groups, and doses... and not shown as a histogram – Figure 5 – also lacking definitions). Supplementary Figure 1 should be corrected.

All figures have been revised according to the reviewers’ comments. Additionally, p-values, n and statistical test have been applied (when relevant) to the figures.

9. Since age is an influencing factor in immune response upon vaccination and might be related to comorbidities, I suggest evaluating association using linear regressions adjusting for age.

Thank you for your suggestion. We have evaluated the influence of age on other variables such as comorbidities by conducting a linear regression analysis adjusted for age. Please refer to the revised Supplementary Table 1 (former Table 2). Our results indicate that, after adjusting for age, the association between the immune response (plasma total Ig anti-spike levels) and the other included variables mostly remained unchanged or was minimally affected, suggesting that age is not a confounding factor in these associations based on our measurements. However, the negative impact of comorbidity on the immune response was no longer found after adjusting for age. We have incorporated these findings into the Results and Discussion sections.

10. State the non-significant p-values in Table 2 to provide a general overview for tendencies.

All the p-values have now been added to Table 2, now referred to as Supplementary Table 1.

11. In which figure are presented results in lines 230-234? Are individuals with two and three doses combined when infection status is evaluated?

There is no figure presenting these results. We have changed the entire Results section including this part and the mentioned results can now be found in Supplementary Table 1 and Figure 2. A dose stratification comparing infection status is illustrated in Figure 3.

12. Has been used the same laboratory assay to quantify antibodies in both serum and saliva? Is this assay validated for both fluids with the same cut-off?

1. Yes, the same assay was used.

2. The IVD approval covers plasma and therefore the assays were validated in-house to be used for measuring in saliva, manuscript in preparation by Boysen O et al.

13. What is the correlation coefficient from the Spearman test between serum and saliva antibodies?

The Spearman correlation coefficient is 0.44. We have decided to revise Figure 5 for better visualization. Please see Figure 6 (former figure 5).

14. Which test and in which Figure/Table can be found the association between IFN- γ and the other factors? (Lines 263-264). In the method section, it is mentioned the evaluation of IFN- γ levels released from CD3+ cells, are these results used to calculate the specific CD4+/8+ IFN- γ levels, or how is the analysis performed??

1. We have added a supplementary Table 4 to support the results/associations.

2. The Quantiferon Monitor belongs to the interferon gamma release assay (IGRA) family of assays. The assays are whole blood (heparinized) assays that use Toll Like Receptor (TLR) 7/8 and T cell receptor (anti-CD3) agonists to stimulate T-cells and monitors the stimulation by quantifying the release of interferon gamma from the cells after approx. 24 hrs incubation of blood tubes at 37C. The tubes contain a serum separator gel and are spun to separate the plasma from the cells IFN-gamma. The Interferon gamma is then measured using an ELISA (QuantIFERON-TB Gold Plus test (QFT-Plus) performed on an automated analyzer (Liaison XL platform, Diasorin)). We used the automated QFT-Plus ELISA (Qiagen, Hilden, Germany) on the Liaison XL instrument. The routine IGRAs are most usually used to evaluate T-cell for specific pathogens, most often TB (Pai M, Denkinger CM, Kik SV, et al. Gamma interferon release assays for detection of Mycobacterium tuberculosis infection. Clin Microbiol Rev 2014; 27(1): 3-20), but also EBV, CMV and so forth. The specific IGRA assays use carefully selected peptides to either stimulate CD4(+8) cells or only CD8. In contrast to the specific IGRAs the Monitor (QFM), assess the overall performance of the innate and adaptive immune system by using both R848 and anti-CD3 antibodies, respectively. This is described in lines 177-191.

15. Have the authors considered the possible spill-over of antibodies from serum into the saliva, indicating the association of these two fluids (lines 291-294)?

The Sarstedt Salivette saliva collection system is a well-established system for collecting saliva and is used for standard collection of saliva for measuring Cortisol in saliva for diagnostic purposes. In research it is also used for collecting saliva for measuring many other substances. As long as the oro-pharyngeal mucosa is intact, leakage has not been a problem for cortisol measurement but in cases where there is oro-pharyngeal inflammation there could be leakage. In our setting, we do not believe that leakage was a problem since none of the participants reported being sick at the time of sample collection or gave any other indication of damage/problem with their oral mucosa.

16. Have the authors considered the possible lack of correlation between IFN- γ and antibodies due to saturation of the serum assay? And the T-cell detection in non-infected individuals due to vaccination? (lines 314-316)

1. It is quite possible that this could be part of the case. However, we performed a correlation test exclusively between the observations with plasma measurements \leq 25000 and IFN-gamma values (leading to 14% of the observations being excluded from the analysis), and they still did not correlate. Therefore, our results cannot be immediately explained by the saturation of the serum assay.

2. Yes, the T-cell assay used detects IFN γ release as a marker for T-cell response to spike protein peptides irrespective of whether the response is caused by exposure to vaccines, SARS-CoV-2 or both.

Minor comments:

17. Provide a reference relating to a) the vaccine roll-out or their implication in disease severity/death decrease (Line 56); b) regarding Greenland population (lines 96, 104, and 105); and c) regarding identification of potential cofounders (lines 179-180).

Relevant references have been added to all three points.

18. The first sentence appears a bit out of context since containment restriction lifts were implemented since transmission, hospital pressure, and disease severity decreased (line 57).

We consider this sentence important as it highlights the difference between the situation Greenland and that of many other countries. We are having trouble understanding why this sentence appears out of context, but if the editors want us to rewrite the sentence please elaborate on the comment.

19. How many COVID-19 cases (lines 64-66)?

21,419 cases per 100,000 inhabitants as of June 2022 – this sentence have been added to the main text (line 67).

20. What were the criteria chosen by the authors for the reinfection definition? Line 206, what was the time between positive tests that 5.7% of participants with more than one positive test (reinfection)?

1. To date, there is no universally accepted definition of COVID-19 reinfection. However, European countries commonly consider a time frame of 45–90 days from a previously confirmed infection to a newly confirmed infection as a reinfection (Source: European Centre for Disease Prevention and Control. Reinfection with SARS-CoV-2: implementation of a surveillance case definition within the EU/EEA. 8 April 2021. ECDC: Stockholm, 2021). To minimize the risk of categorizing an 'old' COVID-19 infection as a reinfection, we opted to define the range up to 90 days.

2. Regarding the section on participants with a reported reinfection, we have now added the median time (4 months, IQR 3-4 months) between positive tests to the text (Line 238).

21. Correct the statistical test used in line 171 since the data analyzed is not paired.

The Wilcoxon-Mann-Whitney test (also known as Mann-Whitney U test) is used for independent, non-paired samples. We therefore believe that this is the correct statistical test for this analysis.

22. Specify the antigen in line 228.

We have specified the antigen to the plasma total S-Ab response.

Reviewer #2 (Remarks to the Author):

Brief summary of the manuscript: Robust immune response to COVID-19 vaccination in Greenlanders – an observational study in an Island population

Møller et al. have analyzed the presence of antibodies against COVID-19 in the Greenland population during the following vaccination between the winters of 2021 and 2022.

Overall impression of the work:

The text would benefit from greater structural organization as well as adding in more contextual information regarding the purpose of the study. Below I recommend some writing improvements for clarity in some sections. The rationale of the study design requires further explanation specifically within the introduction section, we see very little information about the COVID-19 epidemiological scenario in Greenland. Given one of the main points of the study is to discuss the unusual transmission pattern observed in Greenland, the lack of specific numbers (e.g., number of cases) compromises contextualization and would be useful in demonstrating the overall impact of the study. The authors provide a sufficient description of statistical methods used to calculate any differences observed between the sample sizes. The sample size is appropriate for the questions being asked. Please, revise the reference list, some of them do not seem to provide enough information or do not follow any official citation styles (e.g., references number 3, 4).

We thank reviewer 2 for their careful consideration of our manuscript and appreciate the opportunity to reflect on and respond to their comments.

We have changed the structure of some parts of the manuscript, the Result section in particular, and have elaborated on the purpose of the study in the Introduction section. Furthermore, we have tried to explain in more detail the rationale for the study design in the Methods section.

All references have been checked and now follow official citation styles.

Specific comments, with recommendations for addressing each comment:

Abstract:

1)Line 29: Include citation, Noahsen et al. Referenced in the introduction line 60

We acknowledge that there are no specific guidelines on whether to use citations in the abstract. However, following best practices in the academic world, we believe that abstracts should generally not include citations unless essential to the text. We view the abstract as a very brief summary of the research work, and relevant references are incorporated in the introductory section instead. However, if the editors wish, we will provide the citations in the abstract.

2)Line 35: Include the number of participants included in the T-cell analyses

The number of participants has now been included in the abstract (line 37).

3)Line 39-41: Include numerical data to demonstrate booster doses and infection enhanced antibody response. Additionally, include data to support the conclusion that comorbidities weakened antibody response.

Thank you for your suggestion; we have included the relevant data to support our conclusions. Please see line 41-42 in the abstract.

4)Line 47: Include citation(s) for other “population groups”

Please see our response to comment 1).

Introduction:

1)Lines 64-66: The following sentences should read: “Despite good vaccination coverage compared to other Arctic regions³, over the course of the pandemic, Greenland experienced a considerable number of COVID-19 cases. However, although few cases were seen prior to the advent of the Omicron (B.1.1.529) ...”

The sentences have been corrected according to the above suggestion.

2)Lines 64-68: It would be interesting to explore further the COVID numbers for Greenland. “Considerable number,” “Few cases,” and “markedly increased” are too vague.

We have added more specific numbers to the section and we hope that it gives a better and more clear understanding of the development of COVID-19 cases in Greenland (line 67-68).

3)Line 71: Remove the word “possible”

The word has been removed.

4)Line 72: With the phrasing “relatively low” there should be a citation including the reference group, meaning include a citation with hospitalization and death rates for population(s) in June of 2022

Citations have been included.

5)Line 73: Include a citation for the number of hospitalizations

Citation has been included.

6)Line 77: Are there any previous studies that demonstrate that Greenlandic or other Inuit populations have different immune responses to vaccines? If so, I would include data from them to emphasize the importance of this study.

Yes, but very few studies. We have added more information + references on the topic in the Introduction section (line 88-90).

7)Line 81: Please give further explanation of what “limited healthcare services” entails, specifically? Additional background information would be beneficial to clarify how Greenland’s population would be impacted differently than other countries would be helpful to strengthen the introduction.

We have specified what limited healthcare services entails in the Introduction section for better clarity (line 73-75).

8)Line 85: Sentence should read “Specific aims were to quantify the COVID-19 vaccine-induced antibody immune response, to analyze the SARS-CoV-2 specific total Ig response in serum, the T-cell response, and total Ig response in mucosa.”

The sentence has been corrected for clarity.

Methods:

1)Line 94-99: How does the fact that most of Greenland’s population is of Inuit descent impact how the pandemic unfolded there? Is that a critical point? If yes, please discuss it further or remove it from the text.

Very good point. We agree that it is not a critical point and have therefore decided to remove from the text.

2)Line 94-99: As discussed in the comments regarding the introduction, including some further details about the healthcare system in Greenland would help contextualize the paper.

Please see our respond to comment 7) under Introduction. Additionally, we have added more information in line 112-116.

3)Line 108-114: I would recommend moving this portion to the supplemental section of the article, it interrupts descriptions of participant demographics. If not, I would create a new section within the methods entitled “Participant enrollment.”

We have moved the section called “Participant enrollment” to the supplemental section as suggested. See Supplementary Note 1.

4)Line 121: replace “enrolment” with enrollment

The misspelling has been corrected.

5)Lines 123 – 124: Please, define if breakthrough infection is infection after 1 dose, 2 doses, or 2 doses plus booster.

It is defined as an infection after 2 doses or more. This definition has been added to the text (line 137).

6)Line 128: Although not in the national system, self-reporting vaccination status could be proved with a personal vaccination card or similar? Please, describe.

Yes, that is typically the case. However, Greenland did not have a system for a personal vaccination card or similar. Vaccinations administered in Greenland were registered in the national health registries, but the vaccines given outside the country were not registered anywhere accessible to this study. Therefore, we chose to rely on self-reporting information as we considered it the most precise estimate of the number and time of booster vaccinations.

7)Line 136: Replace “Salivette Cotisol” with Salivette Cortisol

The misspelling has been corrected.

8)Line 136: Include the company and cat. #

Salivette® Cortisol, (Sarstedt, Nümbrecht Germany) Cat no 51.1534.500 8. This has been added to the Methods section – sample collection and handling.

9)Line 140: This statement contradicts the statement made in Line 132, which states all participants had blood drawn 12 months after their first dose. In line 140 however, you say “another subset of participants had additional blood samples drawn at 12 months after the first vaccine.” I would suggest clarifying the participants in each experimental group.

We have rephrased the sentence to provide better clarity of the method. Please see the revised text (line 145-152).

10)Line 142: Rearrange the sentence to say, “All samples were shipped frozen (-20 oC) to Denmark within four hours after processing for final laboratory analysis.”

The sentence has been rephrased.

11)Line 143: Include the company and cat. #

1. QuantiFERON SARS-CoV-2 Starter Pack Cat. No. / ID: 626715

2. QuantiFERON Monitor (QFM) Cat. No. / ID: 0650-0101

This has been added to the Methods section – sample collection and handling.

12)Line 146: Replace the word “after” with “of”

The sentence has been corrected.

13)Line 152: Explain how the cut-off index is calculated.

The Elecsys measures the N-Ab using ECLIA. The cutoff was chosen to give a specificity $\geq 99,7\%$ for positives, a sensitivity $\geq 92,7\%$ for having been infected according to (Whitaker HJ, Gower C, Otter AD, et al. Nucleocapsid antibody positivity as a marker of past SARS-CoV-2 infection in population serosurveillance studies: impact of variant, vaccination, and choice of assay cut-off. 2021) and verified in our lab (raw data from (Harritshoj LH, Gybel-Brask M, Afzal S, et al. Comparison of 16 Serological SARS-CoV-2 Immunoassays in 16 Clinical Laboratories. J Clin Microbiol 2021; 59)). The high specificity was important for not overestimating the number of true positives in the beginning of the pandemic when the prevalence of individuals who had been infected was (very) low. However, it was also important to set the cutoff so that there were not too many false negative tests. Thus, the cutoff is the chosen balance between these two concerns given the conditions at the time of the study.

14)Line 155-156: Can you explain further the difference between “insufficient (non-protective)” and “not considered sufficient to neutralize.”

It is the same, we have corrected it and made it uniformed by only using non-protective.

15)Line 160: Include a cat. #

LAISON QuantiFERON-TB Gold Plus, Diasorin, Saluggia, Italy Cat No / ID #3110. This has been added to the Methods section - Laboratory tests.

16)Line 162: The end of the sentence is missing a period after control

It has been corrected.

17)Line 167: One or two tailed?

The QuantiFERON results were based on IFN- γ intervals and can thus be perceived as two tailed.

18)Line 177: This is the first time that the impact of ethnicity is mentioned in the text. If that should be discussed, it should be better explored in the introduction section.

Please see our response to comment 6) under Introduction.

19)Line 180: Include a citation to provide basis for the statement “based on existing literature”

Relevant citations have been added to the text.

Results:

1)Line 195: I would advise re-writing the section “study population and descriptive statistics” for clarity. The current paragraphs are confusing.

We have tried to rewrite this section and removed the title ‘Study population and descriptive statistics’, which hopefully appear less confusing.

2)Line 198: Include that it was a 12-month follow-up after their second vaccine dose.

This point has now been included in the text.

3)Line 198: Include the samples who gave saliva samples, are these the same 323 individuals who gave blood samples 12-months after their first vaccine.

Among the 323 individuals who completed the follow-up visit, 107 of them also contributed a saliva sample. These numbers are referenced in the text (lines 222).

4)Line 204: What was the distribution of Moderna and Pfizer vaccines of the people who returned for their 12-month follow-up? What booster did they receive?

97 % was vaccinated with the Moderna vaccine at the follow-up visit and all received booster doses homologous to the primary vaccine. This information has been included in the text (line 225).

5)Line 208: replace “covid-19” with COVID-19

It has been corrected.

6)Lines 212-213: Combine these two sentences for clarity: “At Time 1, 95.6% of participants demonstrated a significant level of total Ig anti-spike antibodies reaching peak levels at 4-6 weeks after the second vaccine dose, with a median of 3147 kBAU/L (IQR 32-25000 kBAU/L)” (Figure 2).

The sentences have been revised. Line 230-232.

7)Line 216: Include a citation indicating that low levels of N-abs indicate low transmission levels (Whitaker et al.)

Reference has been included.

8)Line 217-221: Reword this paragraph for clarity, you first say that antibody levels were stable and then you say that median at Time 2 was five times greater than at Time 1.

We have rewritten and reorganized the section, which we hope provides better clarity of the results.

9)Line 218-219: Is that the median for all timepoints? Are you comparing the two peak Ig anti-spike levels from Time 1 and 2?

Yes, that is the median for all the measurements of the plasma total S-Abs at Time 2, compared to the median for Time 1. Please see the revised section (line 235).

10)Line 220-221: Indicate if these patients had covid (self-reported or officially notified).

This information (self-reported) has been added to the text.

11)Line 225: This is the first time the Charlson comorbidity index is mentioned in the text (besides being added to the epidemiological characterization table). Please justify why this is important and which comorbidities are being considered, include this information within the methods section

A definition of the Charlson Comorbidity Index score (CCI) have been included in the Methods sections under statistical analysis with relevant references stating its importance.

CCI does not directly address those comorbidities that mainly increase the risk of severe COVID-19 infection but this is an often use measure for comorbidity. Therefore, it was important to evaluate the immune response over levels of comorbidities expressed in CCI.

12)Line 227: Please include data to justify this claim.

We have referred to Supplementary Table 1 in the text as this includes the relevant data.

13)Lines 240 – 246: The data presented here is not relevant and is not even mentioned in the discussion section.

We have incorporated the mentioned data into the Results section under the heading 'Independent factors influencing the plasma antibody response.' Additionally, we have provided a detailed discussion of these results in the Discussion section. We believe that these results are relevant to the study and have therefore decided not to remove them from the text.

14) Lines 257-261: Is measuring IFN- γ enough to be classified as a T-cell mediated immune response? The IFN- γ is an important aspect of the cellular response, however other parameters should be included to call it a T-cell mediated response.

We agree that IFN-gamma is a very general measurement parameter, but in our study, it is specific to T-cell activity as we stimulate with specific linear peptide epitopes and then measure IFN-gamma release after 24 hrs, which becomes an expression of T-cell activity, as in the SARS-CoV-2-specific T-cell assay. In the QuantiFERON monitor (QFM) assay, the blood T-cells are stimulated by a Toll-like receptor 7 ligand and anti-CD3 (the T-cell receptor) target T-cells, making the release of IFN-gamma specific to T-cells. We have described this in the Methods section under Laboratory tests.

Discussion:

1) Line 267: We cannot say that the experimental strategy and results presented in this paper are enough to call it an overview of the cellular response.

We have rephrased the sentence, so it now reads: "In this study, we present estimates of the effects on the humoral and cellular antibody response.."

2) Lines 272-273: All results were presented in months after the first dose, but here you are referencing 4-6 weeks after the second dose. I believe that these timepoints should be clarified or edited within the figures, methods, and discussion sections for maximum clarity.

We agree that the study design, and more specifically the time variable, requires a better definition. Reviewer 1 had a similar comment (no. 4), and we have given the following respond:

Originally, we designed the study with inspiration from a similar Danish study with which we aimed to collaborate, hoping to include additional sample collection time points, i.e., between the first and second doses. However, due to a lack of funding, this was not possible. Consequently, the strategy and design of the study, particularly concerning sample collection time points, underwent changes during the study timeline. Given the current situation, we have decided to present the results in terms of time since the second vaccine, as this aligns with our inclusion criteria and provides a better overview of our design and results.

Therefore, we have corrected the definition of the time variable in the text and all relevant tables and figures.

3) Line 279: The whole purpose of the study is only mentioned clearly in this section. Although the authors do discuss the importance of studying this specific population, this should be hinted at in the introduction more clearly. Specifically, the sentence on Line 280, beginning with, "In addition, a specific genetic trait" must be included within the introduction as basis for conducting the study and understanding the study's importance. The more details you can include regarding this, the stronger the paper.

We agree that this should be highlighted in the Introduction, and have therefore added a section on the impact of Inuit ethnicity on vaccine response (line 88-92).

4)Line 304: You include two claims that are contrary to previous literature, do you have any hypotheses for why this may be? If so, include reasonings or citations for these beliefs.

We do not have a clear explanation for why our results are different from previous research. Therefore, we have decided not to propose a hypothesis.

5)Line 307: Were most of these infections between doses or shortly after receiving their second dose? I would caution against claiming that the vaccination series has limited protection against infection.

Some infections were between doses and some were not. We have changed the wording in the text regarding the vaccine protection against infection (line 338-339).

6)Line 316: Do you have a citation to justify that cross-reactivity may be responsible for the T-cell response?

Yes, the relevant citation has been added to the text.

7)Line 318-342: Is this section being published along with the paper? If not, discussing travel restrictions and border closure is a crucial point to explain the numbers registered in Greenland and must be mentioned in detail in the introduction and/or discussion sections.

The whole section is being published along with the paper. Therefore, we have chosen not to elaborate further on these details in either the Introduction or Discussion section.

Figures and tables:

1)Table 1: I would recommend table one to have only the main demographic information (age, sex, vaccine type, presence of co-morbidities). Some aspects are not discussed in the paper such as smoking and alcohol habits. Data should be transferred to a supplementary table or removed.

Thank you for this suggestion. We agree, and have therefore chosen to remove demographic information (smoking habits, alcohol habits, and BMI) that is not addressed in our discussion.

2)Table 2: Include on the statistically significant p-values for all rows in the table

All p-values have been added to Table 2, now referred to as Supplementary Table 1.

3)Figure 1: I think this could be moved to the supplementary material, as it does not demonstrate any “new” data that you have produced.

Thank you for this suggestion. We agree and have moved Figure 1 to supplementary material as well as combining it with the previous supplementary Figure 1 for a better overview of the study design. Please see the revised Supplementary Figure 1.

4)Figure 2: As mentioned above, switching between months after the first dose and weeks after the second dose is confusing. I would suggest potentially changing the axes titles for greater clarity.

Please see our respond to your comment 2) under Discussion.

5)Figure 2: Why are there colored dots (green for Time 1 and red for Time 2) and gray dots? Do they demonstrate different participants? The green and red dots also are larger than the gray dots, is this purposeful? If so, a legend is required to distinguish these differences.

This figure has been revised for maximum clarity. Please see Figure 1.

6)Figure 3: See comments above regarding the Time 1 and Time 2 and axes labels using weeks since second dose.

Please see our respond to your comment 2) under Discussion.

7)Figure 3: Why is the box for Time 1 Inuit participants not encapsulating most of the data points? Additionally, for why is the box between weeks 20-40 whereas Mixed and Non-Inuit populations have most data points between 0-20 weeks.

The figure has been corrected to ensure that the box encapsulates the majority of data points and is properly positioned according to the time on the x-axis. Please see Figure 1.

8)Figure 4: Bars in this figure are too big, I would suggest adjusting for a better visual impression.

The figure has been revised and we hope that the new figure gives a better visual impression. Please see Figure 3 (former figure 4).

9)Figure 4: There are no points, just a line for the first column titled "2 doses + no SARS-CoV-2 infection", was there only one participant? If so, I would remove this column from the comparison.

Given that the group '2 doses + no SARS-CoV-2 infection' had only two participants, we excluded this group from the figure to enhance visual clarity, as recommended. We believe that this exclusion does not significantly impact the overall visualization and interpretation of our results. See Figure 3 (former figure 4).

10)Figure 5: I would adjust how this figure is formatted. It is a bit complicated; I would begin by including the spearman correlation values on the graphs. I would also include the bar graphs on one row and the correlation graphs on another with axes titles.

As this figure appears confusing we have chosen to create better plots illustrating the correlation between the different outcomes. Please refer to Figure 6 (former figure 5).

11)Supplementary Figure 1: I still believe that the axes would be clearer if they were marked regarding weeks/months since the first dose. However, if you would like to keep them labelled as is, I would recommend including this figure in the main text.

Please also see our respond to your comment 3) in this section. We have chosen to combine this figure with the original Figure 1. Please see the revised Supplementary Figure 1.

12)Supplementary Figure 2: Include number of participants in each cohort in the methods section.

The number of participants in each cohort is provided in the Results section and has also been included in the revised Supplementary Figure 2. We do not believe it is necessary to repeat these numbers in the Methods section.

13)Overall: Just as a general note, I would recommend changing the colors for this figure. People who are colorblind are likely unable to distinguish between red and green, I would suggest using something like blue and yellow.

Very good point. We have changed the colors (blue/yellow/grey) in all figures.

14)Overall, for all figures where comparisons are made, please include either asterisks or p-values to visualize the significant comparisons. Also include details of the statistical analyses done for each figure.

We have included p-values, asterisks and/or statistical analyses to all figures when relevant.

15)Overall: Include (n=) values within your figures to help provide clarity on how many people are included in each analysis.

N= values have been included within all figures and tables.

Reviewer #3 (Remarks to the Author):

In this study Møller et al provide a in interesting overview of humoral and cellular immunity in Greenland populations. Given the peculiarity of Greenland geography and ethnic makeup this study is of great interest. I have a few suggestions that could significantly improve the manuscript.

We thank reviewer 3 for their careful consideration of our manuscript and appreciate the opportunity to reflect on and respond to their comments.

Major:

1) The authors discuss the importance of the analysis in an Inuit populations and the reviewer agrees this is most novel and important aspect of the manuscript. The authors should try to expand this point by showing:

- A direct comparison between Inuit and non-Inuit
We agree that a direct comparison between Inuit and non-Inuit is important. We believe that this comparison has been made in regards to the plasma total S-Ab response stated in the Result section line 262 (referring to data in Supplementary Table 1+2) and illustrated in Figure 1+2. However, we have not examined if Inuit/non-Inuit ethnicity was associated with either the oral mucosal antibody response or the specific SARS-CoV-2 T-cell response. We have included supplementary tables presenting linear regression analyses for the oral mucosal antibody response (Supplementary Table 3), cellular response (Supplementary Table 4), and their associations with included variables, such as Inuit ethnicity. Additionally, we illustrate the comparison between Inuit ethnicity groups for both saliva antibody responses and T-cell response in Figure 4 and 5.

- Data on mucosal antibodies
Please see the above comment.

- Data on cellular responses
Please see the above comment.

2) Data that is discussed in the results is not always shown in figures, this makes the data more difficult to interpret. Specifically, lines 249-254 and lines 258-261. Figure 5 probably contains some of this data (correlation between serum and mucosal IgG) but it is really hard to read. Axes should be labeled, and plots separated for better understanding, the p values and R values should be placed inside the plot to aid interpretation. The significance of the bar plots is unclear to me.

We agree that this plot is too difficult to read and we have therefore decided to revise the plot. Please see Figure 6 (former figure 5).

3) The most relevant data presented in table 2 and sup table 1 should be shown in form of a figure (at least the data mentioned in the results) so it is easier to understand. Especially the data on the relationship with age and antibody levels would be interesting to see, have the authors done a similar analysis for the cellular response?

Thank you for your suggestion. We have updated our figures to illustrate the relationship between age and both plasma and oral mucosal antibody responses, as well as the cellular response (Figure 1+4+5). Additionally, we have illustrated the results from Table 1 (now referred to as Supplementary Table 1) in form of forest plots – please see Figure 2.

Given the relatively small size of our study population, creating figures with stratification or grouping of multiple variables does not provide a clear illustration. Therefore, we have opted to concentrate on the most relevant and statistically significant influencing variables (age, Inuit ethnicity, SARS-CoV-2 infection/booster vaccination) in our figures.

Minor:

- 1) Quite obviously it is impossible to judge humoral immunity 12 months after vaccination when 81% of the individuals received a booster and at least 65% seem to have had a breakthrough infection. Infection especially greatly influences the antibody levels at Time2 (Figure 4). This is noted in the discussion, but the authors should make this point much clearer throughout the manuscript and mention this whenever they draw conclusions that antibody levels at 12 months are “stable” for example line 218. The authors could also show the relationship between the time since booster/infection and antibody levels, either with a correlation of by binning patients into early and late.

We acknowledge the challenge of assessing the antibody response at 12 months due to the impact of breakthrough infections and booster doses. To address this concern, we have revised the wording in the Results section regarding the influence of infection and booster vaccination on antibody levels. We believe that these changes enhance the clarity of our results. Line 236-238.

We shown the relationship between time since booster dose and plasma S-Ab levels in Supplementary Table 1.

- 2) The 3 month and the 12 month time points actually comprise a fairly wide range of timepoints (figure 2) this should be stated more clearly in the text when the study design is presented

We have added a comment on our timepoints and study design in the Methods section (line 145-148). Furthermore, we would like to refer to Supplementary Figure 1, which has been revised and hopefully provides a better understanding of the study design.

- 3) Line 217-218: “At Time 2, 2.2% of participants had non-significant levels of total Ig anti-spike antibodies, while 97.8% had significant levels.” Please clarify significant compared to what, baseline?

Non-significant/protective and significant/protective anti-spike antibody levels were defined based on the cut-off level of 300, as stated in the Methods section.

Reviewers' comments:

Reviewer #1 (Remarks to the Author):

The authors have made substantial changes that have significantly improved the manuscript. The quality of the figures is much better now, and all the results are reported in a Figure or Table. I acknowledge all the work done by the authors modifying the manuscript. Most of my comments were addressed; however, I have some extra minor comments:

1. Line 73, as healthcare professionals are mentioned here, substitute workers in line 63 for professionals for consistency.
2. Line 163, include "Spike" before glycoprotein.
3. Sentence covering lines 236-238, would it be a better fit in the discussion?
4. Consider including in Table 1 the number of individuals with previous SARS-CoV-2 infection (mentioned in lines 239 and 240).
5. Paragraph starting in line 252, although p-values can be found in the respective table, report them in the text as it is done in the next paragraph.
6. Line 255, is 1% (95% CI 0-2%) correct? I cannot find these values in the Supp. Table 1.
7. Line 259, correct BNTb2.
8. Line 265, refer to Supplementary Table 1 too.
9. Supplementary Table 2 is not referred to within the text.
10. Supplementary Figure 1: Mention Top and Bottom to ease the reader the understanding of the figure.
11. Supplementary Table 4 and legend of Figure 2: When it is mentioned log, is it log₂ or log₁₀?
12. I suggest looking at the order of the data presented within the text to improve the flow of the text as sometimes the results jump from one figure to a previous one.

The following comments are a continuation of the previous revision round:

15. Regarding the spill-over of antibodies from serum into saliva. The authors are correct about any direct leakage of antibodies from serum due to any damage/problem in the oral mucosa of the participants. I highlight this comment because other studies have shown the transudation of circulating IgG into saliva in unaltered oral mucosa (<https://doi.org/10.1038/s41598-022-12869-z> and PMC1457446) that would benefit from mentioning. No cut-off is presented for saliva levels.
16. Regarding the correlation between IFN-gamma and antibodies. 1. Could there be a lack of correlation due to the different antigens employed in the assays? 2. Line 346, are the authors referring to reactive T-cells in individuals with no previous infection and vaccination or only the first one? If the authors refer to no infection and no vaccination, both should be mentioned.

Reviewer #2 (Remarks to the Author):

What are the major claims of the paper?

They found that the study population had a robust antibody response following vaccination and that the booster had an additive effect on it, as expected.

Are they novel and will they be of interest to others in the community and the wider field?

The premise is interesting since they are looking at a population that has unique immunological aspects, however those aspects were not properly assessed or discussed within the manuscript, which leave us with an antibody assessment after primary and booster vaccinations (we have had several of those in the past few years since the pandemic and vaccination started).

If the conclusions are not original, it would be helpful if you could provide relevant references.

Since 2021, several papers have assessed antibody levels after primary and booster vaccinations, with or without breakthrough infections. Some examples: [10.3389/fpubh.2021.778243](https://pubmed.ncbi.nlm.nih.gov/358243/), [10.1126/sciimmunol.add4853](https://pubmed.ncbi.nlm.nih.gov/358243/), [10.1002/jmv.27420](https://pubmed.ncbi.nlm.nih.gov/358243/), [10.1128/spectrum.02026-22](https://pubmed.ncbi.nlm.nih.gov/358243/), doi.org/10.1038/s41467-022-32254-8

Is the work convincing, and if not, what further evidence would be required to strengthen the conclusions?

An evaluation of the genetic traits that characterize the Inuit population would bring something new to the manuscript. As it is now, it is just an antibody assessment after primary and booster vaccinations.

Summary of the manuscript: Robust immune response to COVID-19 vaccination in Greenlanders – an observational study in an Island population

Møller et al. have analyzed the presence of antibodies against COVID-19 in the Greenland population during the following vaccination between the winters of 2021 and 2022.

Overall impression of the work:

This paper contains interesting data. However, it is not always clear what the authors are trying to conclude, and the overall message of the paper is lost. The paper would benefit from greater organization and further explanation of why this work is important to not only Greenland's population but other isolated countries when facing a pandemic.

Specific comments, with recommendations for addressing each comment:

Abstract:

Line 39: Remove total

Line 40: Remove continued, as it implies that all patients were included in both Time 1 and Time 2

Line 42: Should say "Total salivary S-Ab levels..."

Introduction:

Line 57: replace "already in" with "by"

Line 60-61: Replace "are little described" to "are not well documented"

Line 67-70: Cases appear to be breakthrough infections. Can you please elaborate more on the topic?

Line 71-74: Rephrase the sentence beginning with "The population of Greenland..." Separate into two sentences potentially one describing increased disease and a second sentence to explain the challenges healthcare access in Greenland for greater clarity.

Lines 79 – 83: paragraph is very confusing. Please, re-write for clarification purposes.

Line 85: Remove "both"

Lines 88 – 90: "Information" about Greenlandic population having different immune response to vaccines in general was added, but the topic remains very vague. What kind of diseases? How does this impact disease epidemiology?

Line 90: Remove "these findings make it interesting" and replace it with "it was important to"

Line 91: what genetic factors? Is this approached in the discussion section?

Line 94-95: End the sentence at "similar isolated areas."

Methods:

Line 106: Include a reference to the table with participant demographics

Line 108-118: I do not believe this section belongs in the methods, it should be condensed and included in the introduction.

You removed the "Inuit descent" factor but still mentions in the introduction section as a major point of discussion in the paper.

The topic named "setting" is written in a non-clear way. Please, re-write for clarity purposes.

Line 114: define health educated staff. You mention that hospital settings might or might not have health educated staff. What does this entail specifically?

Line 118: is the fact that Greenland health system is tax-financed relevant?

Line 125-126: End the sentence "before enrollment." Remove everything following.

Line 127: Change "enrolment" to "enrollment"

132-133: Inuit descent is still being approached although the authors stated in the rebuttal letter that the aspect would be removed since they decided to not discuss descent as a relevant aspect in immune responses.

Line 138-142: Move the sentences beginning with "Due to limited resources..." to the limitation section of the paper.

Line 145: Replace "was designed to collect" to "collected"

Lines 145 – 146: Where are the boosters in this timeline? It is important to include this information.

Line 147-152: This should also be moved to the limitations section of the paper.

Line 152: Minor subgroup constituted of how many people?

Line 161: Replace "into" with "in"

Line 183: Should "Quantiferon" be "QuantiFERON?"

Line 184: The sentence should read "a negative control and a positive control..."

Line 187: The sentence should read "The QuantiFERON monitor (QFM) is a one tube..."

Line 195: Take out the comparisons between vaccine type, they cannot be statistically significant since there is a large difference in sample sizes.

Line 199: italicize "a priori"

Results:

Line 220: As referenced above remove direct comparisons between the two different vaccines.

Line 223: might be an interesting point discussing why such a big part of the group was lost during the follow-up.

Line 233: is it safe to say that the antibodies are protective? Are you really assessing a functional aspect here? If it is just a cut-off established by the methodology, please discuss it further.

Line 238: the aspects regarding breakthrough and reinfections are important. Even though the vaccine did not protect them of being infected, it did confer some clinical protection.

Line 252: Remove "significantly" and replace it with 21% (95% CI 7-33%) ..."

Lines 258-261: Remove comparisons between vaccine groups, they cannot be compared based on the difference in sample size.

Line 271: usually the journal asks to avoid "data not shown". Instead, please provide the necessary data to claim any conclusions or remove the sentence/data.

Line 279: Put the figures in order as they appear in the text

Discussion:

Line 312: Was this genetic trait investigated in the study population? According to the authors, said trait is present among a small proportion of them, but it was not assessed in this specific study participants. The phrasing led us to believe that this was a factor.

It might be the first study in an Inuit population but none of the characterizing aspects of said population were assessed.

Line 318: authors claim that mucosal immune responses are not well explored in the SARS-CoV2 context, but a quick search in PubMed results in over 600 published papers.

Line 319: Need to include a citation for the claim "respiratory infections is well characterized"

The early booster generating lower antibody responses mentioned in the results section was not discussed. It's an important point and said results could have a direct impact on public health measures.

Line 329: "Contrary to other findings" needs a citation

Line 346: Were there any age differences in people with cross-reactivity?

Lower antibody responses according to vaccine type (Moderna or Pfizer) at time 1 was also

mentioned in the results section and not discussed.

Line 370: Replace "with" to "to"

Line 371: Why wouldn't they? Please, elaborate.

Line 383-390: Include limitations discussed in the methods section in this paragraph

Line 393-394: "S-Ab responses declined with increasing age." While that is true and the paper shows that this is not a complete thought or conclusion, please expand on why this finding is important.

Figures and tables:

For figure legends change P-values to p-values

Table 1: Change the table description to say "(mRNA-1273 or BNT162b2)"

Figure 1: Why are there grid marks in 1b but not for 1a?

Figure 1: Include statistical significance markers on the graphs, using either their p-values or * symbols between the groups you are comparing

Figure 2a: Cannot compare previous infection group with people who have not previously been infected due to the large difference in sample sizes

Figure 2a: Cannot compare Moderna versus Pfizer vaccine recipients as there is a large difference in sample size

Figure 3: What are the p-values on the graphs referencing, which groups are you comparing?

Reviewer #3 (Remarks to the Author):

The authors have addressed my concerns.

Rebuttal Letter – second revision

Reviewers' comments:

Reviewer #1 (Remarks to the Author):

The authors have made substantial changes that have significantly improved the manuscript. The quality of the figures is much better now, and all the results are reported in a Figure or Table. I acknowledge all the work done by the authors modifying the manuscript.

Thank you very much. We appreciate the additional feedback on our manuscript.

Most of my comments were addressed; however, I have some extra minor comments:

1. Line 73, as healthcare professionals are mentioned here, substitute workers in line 63 for professionals for consistency.

We agree, and the sentence has been corrected as suggested.

2. Line 163, include "Spike" before glycoprotein.

'Spike' has been added to the sentence.

3. Sentence covering lines 236-238, would it be a better fit in the discussion?

We agree that the sentence could be perceived as a discussion rather than a result. We have rephrased the sentence so that it now only appears as a result. Furthermore, we have discussed the significant additive effect of previous SARS-CoV-2 infection on the antibody levels at Time 2. Please see lines 311-314 in the discussion section.

4. Consider including in Table 1 the number of individuals with previous SARS-CoV-2 infection (mentioned in lines 239 and 240).

Thank you for the suggestion. However, the numbers are already mentioned in Supplementary Table 2. If you prefer them reported in Table 1 instead we can move the numbers from Supplementary Table 2.

5. Paragraph starting in line 252, although p-values can be found in the respective table, report them in the text as it is done in the next paragraph.

Relevant p-values (from supplementary information Table 1) have been added to the mentioned paragraph.

6. Line 255, is 1% (95% CI 0-2%) correct? I cannot find these values in the Supp. Table 1.

Yes, it is correct. The estimate can be found in Supp. Table 1 under *Interval (weeks) since booster vaccination (3rd dose)*, Model 1^a at Time 2. However, for better understanding, the estimate has been presented in percentages. Alternatively, we could present the p-values instead if it offers better clarity.

7. Line 259, correct BNTb2.

The misspelling has been corrected to BNT162b2.

8. Line 265, refer to Supplementary Table 1 too.

We now also refer to Supplementary Table 1 in the text as suggested.

9. Supplementary Table 2 is not referred to within the text.

Thank you for bringing this to our attention. Line 244 in the results section now includes a reference to Supplementary Table 2 (changed to Supplementary Table 1):

“Over time, there was an increase in overall antibody levels (Supplementary Table 1).”

10. Supplementary Figure 1: Mention Top and Bottom to ease the reader the understanding of the figure.

This has been added to the figure legend.

11. Supplementary Table 4 and legend of Figure 2: When it is mentioned log, is it log₂ or log₁₀?

We have included in the relevant figure and table legends, as well as in the statistical analysis paragraph, that the data is presented as log₁₀.

12. I suggest looking at the order of the data presented within the text to improve the flow of the text as sometimes the results jump from one figure to a previous one.

Thank you for bringing this to our attention. We have changed the order of some of the results according to the table and figure numbers to improve the flow of the text.

The following comments are a continuation of the previous revision round:

15. Regarding the spill-over of antibodies from serum into saliva. The authors are correct about any direct leakage of antibodies from serum due to any damage/problem in the oral mucosa of the participants. I highlight this comment because other studies have shown the transudation of circulating IgG into saliva in unaltered oral mucosa (<https://doi.org/10.1038/s41598-022-12869-z> and PMC1457446) that would benefit from mentioning. No cut-off is presented for saliva levels.

Thank you for the additional comment on this subject. We acknowledge the findings stated in the provided references and have added to the discussion by mentioning the possibility that such a spill-over of antibodies could have affected our outcome.

Additionally, it is correct that we do not provide a cut-off value for saliva levels of total Ig, as, to our knowledge, there have been no studies that have determined a cut-off level. However, as previously mentioned, the applied assays were validated in-house for measuring saliva, as described in the manuscript in preparation by Boysen O et al. The preliminary cut-off values are:

Saliva N-Ab: Cutoff <0.4 COI

Saliva S-Ab: Cutoff <0.4 KBAU/L

We have chosen not to incorporate these cut-off values in the paper due to it being unpublished information.

16. Regarding the correlation between IFN-gamma and antibodies. 1. Could there be a lack of correlation due to the different antigens employed in the assays? 2. Line 346, are the authors referring to reactive T-cells in individuals with no previous infection and vaccination or only the first one? If the authors refer to no infection and no vaccination, both should be mentioned.

1. Yes, it could be the case that the assays used and the peptides they use for T-cell activation could have influenced our results. We have added these thoughts to the discussion:

“The lack of correlation could be due to the assays used and the peptides they use for T-cell activation; however, it is not entirely clear why our results differ from previous findings.”

2. We are referring to unvaccinated individuals with no previous infection. This has been added to the text, so it now reads:

“Notably, even in unvaccinated individuals with no previous infection, SARS-CoV-2 reactive T-cells have been detected, which may indicate cross-reactivity with other human coronaviruses.”

Reviewer #2 (Remarks to the Author):

The premise is interesting since they are looking at a population that has unique immunological aspects, however those aspects were not properly assessed or discussed within the manuscript, which leave us with an antibody assessment after primary and booster vaccinations (we have had several of those in the past few years since the pandemic and vaccination started). Since 2021, several papers have assessed antibody levels after primary and booster vaccinations, with or without breakthrough infections. Some examples: [10.3389/fpubh.2021.778243](https://pubmed.ncbi.nlm.nih.gov/358243/), [10.1126/sciimmunol.add4853](https://pubmed.ncbi.nlm.nih.gov/358243/), [10.1002/jmv.27420](https://pubmed.ncbi.nlm.nih.gov/358243/), [10.1128/spectrum.02026-22](https://pubmed.ncbi.nlm.nih.gov/358243/), doi.org/10.1038/s41467-022-32254-8

Is the work convincing, and if not, what further evidence would be required to strengthen the conclusions?

Thank you for your additional feedback on our manuscript. We acknowledge that many papers have assessed the humoral and cellular COVID-19 vaccine-induced response after booster vaccinations since the introduction of the vaccines. However, we believe that the unique context of our study is its strength and should be emphasized.

A study on the COVID-19 vaccine-induced immune response in an Inuit population, (which as the reviewer points out, has unique immunological aspects) has not been conducted before. Furthermore, the population of Greenland, which is 89.7% Inuit, is part of the Arctic, which is an area with unique environmental and social challenges. The challenges, which are also highlighted in the manuscript, consist of a limited healthcare system with large distances between healthcare facilities, and healthcare professionals, and a lack of specialized equipment such as respirators and intensive care units. Additionally, the population is at high risk of severe COVID-19 disease due to the high number of chronic diseases such as obesity, diabetes, and cardiovascular disease as well

as poor living conditions (i.e. overcrowding). To prevent the spread of the SARS-CoV-2 virus, the Greenlandic authorities implemented early in the pandemic strict travel measures such as border controls, mandatory quarantine, and testing of travelers to prevent the importation of the virus. Additionally, to prevent community transmission, they temporarily closed public spaces, workplaces, schools, and daycare centers. However, these restrictions were lifted in the spring/summer of 2020 because the containment strategy was successful in preventing the importation of the virus. This gave the authorities ample time to vaccinate the population and protect them from severe COVID-19 disease.

It was important that the Greenlandic population, alongside other Arctic communities, had effective measures in place during the pandemic to safeguard their vulnerable populations. Among the most crucial public health interventions was the deployment of vaccines, which had demonstrated high efficacy in clinical trials. However, given the lack of scientific knowledge regarding how COVID-19 vaccines would perform in this isolated, potentially immunologically distinct, and particularly vulnerable population, Greenlandic authorities, like many other minority populations, had to rely on the available evidence at the time. Consequently, they initiated vaccination efforts to protect the population from severe COVID-19 disease and death. Distributing the vaccines was a challenging task, as a significant portion of the population was residing in smaller towns and settlements scattered along the coast. Healthcare workers carried the vaccines in large freezers on ships and traveled to each town or settlement where the population had gathered in local community halls or gyms to get vaccinated. In the northern regions, some settlements were only accessible by dog sled.

Given the above-mentioned circumstances, we found it important to investigate the specific immunological vaccine response in the Greenlandic Inuit population as a proxy of vaccine effectiveness, as they constituted the primary defense against the COVID-19 pandemic in Greenland.

What are the major claims of the paper? They found that the study population had a robust antibody response following vaccination and that the booster had an additive effect on it, as expected.

As the study progressed, so did the pandemic, with community transmission becoming a reality in Greenland in the autumn of 2021. Despite this, as emphasized in the manuscript, only 27 individuals were hospitalized (none requiring intensive care or respiratory assistance), and only 12 deaths were reported, which stands in contrast to the impact of the pandemic in the majority of other countries. By the time community transmission began, many Greenlanders had already received primary vaccination. Concurrently, globally and in Greenland, milder SARS-CoV-2 variants (Delta and Omicron) had become dominant. Consequently, we concluded that both factors—effective vaccine immune responses and the prevalence of milder variants—likely contributed to the remarkably mild course of infection in Greenlanders.

Are they (the conclusions) novel and will they be of interest to others in the community and the wider field? If the conclusions are not original, it would be helpful if you could provide relevant references.

We believe that although COVID-19 is no longer considered a critical illness, understanding how the unique population of Greenland, with its specific genetics and environmental conditions, responded to the vaccines remains important. This knowledge is essential for preparing Greenland and other minority populations with distinct pandemic experiences for future epidemics. This perspective has been incorporated into the conclusion to underscore the relevance of our original findings to others in the community and wider field.

This paper contains interesting data. However, it is not always clear what the authors are trying to conclude, and the overall message of the paper is lost. The paper would benefit from greater organization and further explanation of why this work is important to not only Greenland's population but other isolated countries when facing a pandemic.

We have provided a more detailed account of the course of infection in Greenland, clarified the specific objectives of the study, and expanded on our conclusions to enhance the study's clarity.

An evaluation of the genetic traits that characterize the Inuit population would bring something new to the manuscript. As it is now, it is just an antibody assessment after primary and booster vaccinations.

We acknowledge that assessing the genetic traits related to COVID-19 disease in the Inuit community would be highly interesting. However, conducting such examinations would be ethically challenging and would require a completely separate study. Moreover, the primary focus of this paper is on analyzing the immune response to the COVID-19 vaccine among Greenlanders, rather than specific genetic traits which is beyond the scope of this study.

Summary of the manuscript: Robust immune response to COVID-19 vaccination in Greenlanders – an observational study in an Island population

Møller et al. have analyzed the presence of antibodies against COVID-19 in the Greenland population during the following vaccination between the winters of 2021 and 2022.

Specific comments, with recommendations for addressing each comment:

Abstract:

Line 39: Remove total

The word 'total' has been removed.

Line 40: Remove continued, as it implies that all patients were included in both Time 1 and Time 2

The word 'continued' has been removed.

Line 42: Should say "Total salivary S-Ab levels..."

We have rephrased the sentence according to the suggestion.

Introduction:

Line 57: replace “already in” with “by”

We have rephrased the sentence according to the suggestion.

Line 60-61: Replace “are little described” to “are not well documented”

We have rephrased the sentence according to the suggestion.

Line 67-70: Cases appear to be breakthrough infections. Can you please elaborate more on the topic?

We cannot from our data determine whether these cases are breakthrough infections. We present the number of cases per 100,000 inhabitants at different time points to illustrate the course of infection in Greenland. These case numbers are irrespective of vaccination status, as our aim is to emphasize that milder variants like the Omicron variant significantly increased case numbers in the country, leading to the onset of community transmission which did not occur until the winter of 2021.

Line 71-74: Rephrase the sentence beginning with “The population of Greenland...” Separate into two sentences potentially one describing increased disease and a second sentence to explain the challenges healthcare access in Greenland for greater clarity.

We have rephrased the sentence according to the suggestion. It now reads:

“The population of Greenland is characterized by high rates of respiratory and chronic disease including obesity, and inadequate housing (i.e. overcrowding). Additionally, logistical barriers to accessing health services exist, such as large distances that can only be covered by flight or boat, limited availability of hospitals in remote regions, lack of healthcare professionals, and specialized equipment like ventilators.”

Lines 79 – 83: paragraph is very confusing. Please, re-write for clarification purposes.

We have rephrased the mentioned paragraph for better clarification. It now reads:

“To minimize the import of SARS-CoV-2 to Greenland, strict travel restrictions such as complete border closure, mandatory quarantine, and testing of all travelers entering Greenland were quickly implemented by the Greenlandic authorities. These measures aimed at delaying the peak of transmission, providing time for the implementation of additional public health interventions, including vaccines. The relatively low numbers of COVID-19 related hospitalizations (27 individuals) and deaths (11 individuals) as of June 2022 suggest, among other mitigating factors, the protective value of vaccine immunization before the occurrence of community transmission.”

Line 85: Remove “both”

The word ‘both’ has been removed.

Lines 88 – 90: “Information” about Greenlandic population having different immune response to

vaccines in general was added, but the topic remains very vague. What kind of diseases? How does this impact disease epidemiology?

1. We have elaborated on the immunogenicity of vaccines in Inuit populations. The paragraph now reads:

“Some studies suggest that the immunogenicity of COVID-19 vaccines can be influenced by ethnicity. Given that previous research has shown differences in vaccine response between Inuit populations and other ethnic groups for a range of other infections such as measles and Haemophilus influenzae type b, it was important to investigate the influence of ethnicity particularly within the Inuit population, on the response to the COVID-19 vaccines. The immunogenicity of vaccines in Inuit populations is not well explored and the reasons for different vaccine responses among Inuits and non-Inuits are unclear. However, these previous studies highlight the need for immunogenicity studies on indigenous populations before generalizing the results of vaccine trials done in predominantly Caucasian populations.”

2. We cannot explain how the COVID-19 vaccines' specific immunogenicity in Inuit populations affects disease epidemiology.

Line 90: Remove “these findings make it interesting” and replace it with “it was important to”

We have rephrased the sentence according to the suggestion.

Line 91: what genetic factors? Is this approached in the discussion section?

We have rephrased the sentence and replaced 'genetic factors' with 'the influence of ethnicity' as this is a more accurate description of the study's aim.

Line 94-95: End the sentence at “similar isolated areas.”

We have rephrased the sentence according to the suggestion.

Methods:

Line 106: Include a reference to the table with participant demographics

We refer to Table 1 in the results section. We do not believe that it should be referred to in the Methods section as well as this would be confusing. However, we have added a short description of the reported data in Table 1 in the Methods section:

“Demographic data were reported according to vaccine type as percentages or medians with interquartile range (IQR). IQR was defined as the range between the 25th percentile (Q1) and the 75th percentile (Q3) of the data when sorted in ascending order.”

Line 108-118: I do not believe this section belongs in the methods, it should be condensed and included in the introduction.

We respectfully disagree with this suggestion. It is quite common to include a 'Settings' paragraph in the Methods section when conducting research in Greenland or similar areas, where the unique settings necessitate additional information.

You removed the “Inuit descent” factor but still mentions in the introduction section as a major point of discussion in the paper.

As requested by the reviewer during the initial manuscript revision, we omitted the description of the Greenlandic population stating that 89% is of Inuit descent. However, we retained the remaining sections discussing Inuit ethnicity throughout the manuscript, as we consider it a crucial aspect to address and one of the primary motivations behind conducting this study. Given that the matter is raised once more, it now seems appropriate to reintroduce the information. Consequently, we have reinstated the description under the "Settings" section.

The topic named “setting” is written in a non-clear way. Please, re-write for clarity purposes.

Thank you for the comment. We have rephrased the paragraph for better clarity.

Line 114: define health educated staff. You mention that hospital settings might or might not have health educated staff. What does this entail specifically?

"Health-educated staff" refers to individuals who have received formal education or training in the field of health. These individuals possess knowledge and skills related to various aspects of healthcare, including medical terminology, patient care, disease prevention, and treatment protocols. They may include healthcare professionals such as MDs, nurses, nurse practitioners, midwives, healthcare assistants, and health aides.

The definition has been added to the text:

“Other towns in each region have healthcare centers, while settlements are equipped with healthcare stations with or without health-educated staff (MDs, nurses, nurse practitioners, midwives, healthcare assistants, and health aides).”

Line 118: is the fact that Greenland health system is tax-financed relevant?

Yes, this information is included to provide a respectful and inclusive description of the Greenlandic healthcare system, considering its history of colonization when Greenland did not have financial or administrative control over its own healthcare system. We believe it is important to acknowledge the financial structure of the healthcare system during the COVID-19 pandemic, where the Greenlandic health authorities were in control and made critical decisions that prevented the devastating impacts of the pandemic.

Line 125-126: End the sentence “before enrollment.” Remove everything following.

We have rephrased the sentence according to the suggestion.

Line 127: Change “enrolment” to “enrollment”

The misspelling has been corrected.

132-133: Inuit descent is still being approached although the authors stated in the rebuttal letter that the aspect would be removed since they decided to not discuss descent as a relevant aspect in immune responses.

Please also see our response above. As requested by the reviewer, we removed the mention of Inuit descent from the "Settings" paragraph. However, we retained this aspect throughout the manuscript because we consider it important to discuss in relation to immune responses and as one of the primary motivations for conducting this study. We firmly believe that emphasizing Inuit ethnicity as a focal point of this research, alongside the unique epidemiological course of the COVID-19 pandemic in Greenland, holds significant importance.

Line 138-142: Move the sentences beginning with "Due to limited resources..." to the limitation section of the paper.

We respectfully disagree with this suggestion as it is historical information regarding the register or lack of register of test results in Greenland and therefore belongs in the Methods section where it explains our data sources. We have already mentioned the limitations of using self-reported data on booster vaccination and test results in the limitations section.

Line 145: Replace "was designed to collect" to "collected"

We have rephrased the sentence according to the suggestion.

Lines 145 – 146: Where are the boosters in this timeline? It is important to include this information.

Please see Supplementary Figure 1 for this information.

Line 147-152: This should also be moved to the limitations section of the paper.

We have already included this information in the limitation section of the paper.

Line 152: Minor subgroup constituted of how many people?

We have reported the number of subgroups in the results section. Please see lines 231-232.

Line 161: Replace "into" with "in"

We have rephrased the sentence according to the suggestion.

Line 183: Should "Quantiferon" be "QuantiFERON"?

Yes, it should be. Thank you for the correction.

Line 184: The sentence should read "a negative control and a positive control..."

The sentence is correct as the PHA in the positive control stands for Phytohemagglutinin. However, we have added an explanation of the abbreviation in the text for clarification.

Line 187: The sentence should read "The QuantiFERON monitor (QFM) is a one tube..."

This has been corrected according to the suggestion.

Line 195: Take out the comparisons between vaccine type, they cannot be statistically significant since there is a large difference in sample sizes.

We agree that there is a large difference in sample size between the two groups when comparing vaccine types. Although the sample sizes are very different, this difference is taken into account in the statistical tests as they account for variations in sample sizes. They compute p-values based on both sample sizes and differences in estimates. We believe that we have used the tests correctly and therefore should not exclude the comparison between the two vaccine types.

Line 199: italicize “a priori”

A priori has been italicized.

Results:

Line 220: As referenced above remove direct comparisons between the two different vaccines.

Please see our reply above.

Line 223: might be an interesting point discussing why such a big part of the group was lost during the follow-up.

We do not believe that a loss to follow-up of 25% is very uncommon in observational studies like this. When compared to other studies on COVID-19 vaccine response, our numbers are quite acceptable. For example:

Søgaard, O. S. et al. Characteristics associated with serological COVID-19 vaccine response and durability in an older population with significant comorbidity: the Danish Nationwide ENFORCE Study. Clin. Microbiol. Infect. 28, (2022). This is based on a large Danish national study, and they report 86.5% and 62.6% follow-up at the study visits 90 and 180 days, respectively.

Fedele G et al. A 12-month follow-up of the immune response to SARS-CoV-2 primary vaccination: evidence from a real-world study. Front Immunol. 2023 Nov 20;14:1272119. doi: 10.3389/fimmu.2023.1272119. PMID: 38077369; PMCID: PMC10698351. A population-based study conducted in Italy with a 47% follow-up rate after 12 months since enrollment.

Line 233: is it safe to say that the antibodies are protective? Are you really assessing a functional aspect here? If it is just a cut-off established by the methodology, please discuss it further.

Protective levels are defined in the Methods section with a cut-off at 300 kBAU/L (based on previous research by Winichakoon, P. et al. Diagnostic performance between in-house and commercial SARS-CoV-2 serological immunoassays including binding-specific antibody and surrogate virus neutralization test (sVNT). Sci. Reports 2022 131 13, 1–9 (2023)). Additionally, a recent study (Feng S et al. Correlates of protection against symptomatic and asymptomatic SARS-CoV-2 infection. Nat Med. 2021 Nov;27(11):2032-2040. doi: 10.1038/s41591-021-01540-1. Epub 2021 Sep 29. PMID: 34588689; PMCID: PMC8604724) found that antibody levels greater than 264 BAU/mL symptomatic SARS-CoV-2 infection. Therefore, we chose to use an approximate cut-off of 300 kBAU/L due to uncertainties in the cut-off estimate.

We acknowledge that cut-off levels representing functional aspects of the antibodies have not yet been determined, and our definition therefore serves as a proxy definition of protection.

Line 238: the aspects regarding breakthrough and reinfections are important. Even though the vaccine did not protect them of being infected, it did confer some clinical protection.

Yes, we agree. For the same reasons we have stated in the Discussion (lines 352-357) :

“Although we found a robust immune response in our study population up to 11 months following primary vaccination, a substantial proportion of participants (74%) had experienced infection between our data collection points. Consistent with other studies, this suggests limited vaccine protection against infection with the Omicron variant, which was dominant in this period. However, it is noteworthy that despite many individuals being infected after vaccination, none reported serious illnesses or required hospitalization. These observations further support the long-term protectiveness of the vaccines against severe disease and death.”

Line 252: Remove “significantly” and replace it with 21% (95% CI 7-33%) ...”

This has been corrected according to the suggestion.

Lines 258-261: Remove comparisons between vaccine groups, they cannot be compared based on the difference in sample size.

Please see our reply above.

Line 271: usually the journal asks to avoid “data not shown”. Instead, please provide the necessary data to claim any conclusions or remove the sentence/data.

As the additional analysis for interaction did not show any significant results, we have decided to remove the sentence as suggested.

Line 279: Put the figures in order as they appear in the text

Thank you for bringing this to our attention. We have changed the order of some of the results according to the table and figure numbers to improve the flow of the text.

Discussion:

Line 312: Was this genetic trait investigated in the study population? According to the authors, said trait is present among a small proportion of them, but it was not assessed in this specific study participants. The phrasing led us to believe that this was a factor.

It might be the first study in an Inuit population but none of the characterizing aspects of said population were assessed.

No, the focus of this study was not to examine the proportion of the IFNAR2 gene defect in our study population. We aimed to determine the COVID-19 vaccine-induced immune response in an Inuit population. The mentioned reference is highlighted in the discussion to provide a genetic perspective on our findings and to discuss the purpose of the study, but not to discuss the IFNAR2 gene defect in itself, as we have no data to support that this gene is involved in COVID-19 immune

response.

Line 318: authors claim that mucosal immune responses are not well explored in the SARS-CoV2 context, but a quick search in PubMed results in over 600 published papers.

We have rephrased the sentence and added relevant references to support the text:

“While the significance of oral mucosal immune responses in preventing respiratory infections is well-recognized³⁰, this is only beginning to be explored in the context of SARS-CoV-2 infection.”

- Escalera, A. *et al.* SARS-CoV-2 infection induces robust mucosal antibody responses in the upper respiratory tract. *iScience* 27, 109210 (2024).
- Havervall, S. *et al.* Anti-Spike Mucosal IgA Protection against SARS-CoV-2 Omicron Infection. *N. Engl. J. Med.* 387, (2022).

Line 319: Need to include a citation for the claim “respiratory infections is well characterized”

We have added a relevant reference as suggested.

The early booster generating lower antibody responses mentioned in the results section was not discussed. It’s an important point and said results could have a direct impact on public health measures.

While this point was reported in the Results section, we found that there was minimal difference between antibody responses, with nearly identical estimates (0.99 and 1.00 after age adjustment) and a non-significant p-value after age adjustment (p=0.652). Therefore, we did not consider this finding significant enough to highlight.

Line 329: “Contrary to other findings” needs a citation

We have added a relevant reference as suggested.

Line 346: Were there any age differences in people with cross-reactivity?

The one study (Moss, P. The T cell immune response against SARS-CoV-2. *Nat. Immunol.* 2022 232 23, 186–193 513 (2022)) that we are referring to says that:

“T cell responses against cross-reactive human coronavirus (HCoV) are generated but are of relatively low magnitude, and their longevity is uncertain, with low frequency in older people.”

As well as:

“It is noteworthy that children and young adults show higher levels of antibody cross-reactivity between HCoVs and SARS-CoV-2, potentially as a result of more recent HCoV infection, and antibodies that can neutralize SARS-CoV-2 are detectable in some children prior to any exposure to SARS-CoV-2.”

Lower antibody responses according to vaccine type (Moderna or Pfizer) at time 1 was also mentioned in the results section and not discussed.

This is not correct as we discuss these results in lines 319-322.

Line 370: Replace “with” to “to”

This has been corrected according to the suggestion.

Line 371: Why wouldn't they? Please, elaborate.

As mentioned in the introduction, previous research has indicated that Inuit populations may have distinct vaccine-induced immune responses to other infections. Therefore, it is important to investigate whether this applies to COVID-19 vaccines. Additionally, as highlighted in both the introduction and discussion, Greenlanders had a mild experience with COVID-19 when community transmission began in the country. Few hospitalizations and only 12 reported deaths nationwide occurred. This was likely due partly to the prevalence of milder variants during the transmission period. However, it is also important to examine the vaccine-induced immune response in Greenlanders, as this may have contributed to the mild course of the pandemic. Despite not hypothesizing that the immune response would be markedly different from other populations, the lack of studies on vaccine immune response in Inuit populations (or the Arctic in general) and the aforementioned circumstances made this exploration important.

Line 383-390: Include limitations discussed in the methods section in this paragraph

We believe that it should be mentioned both in the methods and discussion to display transparency regarding our study methods.

Line 393-394: “S-Ab responses declined with increasing age.” While that is true and the paper shows that this is not a complete thought or conclusion, please expand on why this finding is important.

This sentence is included to demonstrate that antibody responses decline with increasing age, similar to findings from studies conducted in other populations. This suggests that age should be taken into consideration in future vaccination campaigns. We have included this argument in the conclusion:

“S-Ab responses declined with increasing age, consistent with findings from other populations, indicating that age should be taken into consideration in future vaccination campaigns.”

Figures and tables:

For figure legends change P-values to p-values

Thank you for your suggestion. However, we have chosen not to change the text because we believe it is grammatically correct to start a sentence with a capital letter after punctuation.

Table 1: Change the table description to say “(mRNA-1273 or BNT162b2)”

This has been changed according to the suggestion.

Figure 1: Why are there grid marks in 1b but not for 1a?

Thank you for bringing this to our attention. We have now added grid marks to Figure 1a as well.

Figure 1: Include statistical significance markers on the graphs, using either their p-values or * symbols between the groups you are comparing

We do not make any comparisons in Figure 1 and we therefore see no need for significance markers on the graph.

Figure 2a: Cannot compare previous infection group with people who have not previously been infected due to the large difference in sample sizes

Please see our reply above concerning comparisons of groups with large differences in sample sizes.

Figure 2a: Cannot compare Moderna versus Pfizer vaccine recipients as there is a large difference in sample size

Please see our reply above.

Figure 3: What are the p-values on the graphs referencing, which groups are you comparing?

We have clarified the p-values in Figure 3 and elaborated on our statistical methods in the text and corresponding figure legend.

“The Kruskal-Wallis test was used to examine the overall differences in antibody and cellular response across different groups categorized by previous SARS-CoV-2 infection and/or a booster dose. Subsequently, pairwise comparisons were performed using Dunn's test, with adjustments made for multiple comparisons through Bonferroni correction.”

Reviewer #3 (Remarks to the Author):

The authors have addressed my concerns.

REVIEWERS' COMMENTS:

Reviewer #1 (Remarks to the Author):

Thank you for addressing all the comments. I have no further additions.

Reviewer #2 (Remarks to the Author):

All comments have been addressed, so we are not requesting any more revisions.